# Dynamic ARDL Simulations Effects of Fiscal Decentralization, Green Technological Innovation, Trade Openness, and Institutional Quality on Environmental Sustainability: Evidence from South Africa

**Maxwell Chukwudi Udeagha \*** and **Nicholas Ngepah**

School of Economics, College of Business and Economics, University of Johannesburg, Johannesburg 2006, South Africa
* Correspondence: maxwelluc@yahoo.com

**Abstract:** Fiscal decentralization and green innovation are important to a country's economic progress, but the externalities of increased pollution as a result of a rise in the energy used and economic growth must not be overlooked. The destruction of the environment presents a serious threat to human existence. South Africa, like several nations, has been working on reducing its dependence on fossil fuels such as coal by utilizing modern energy-efficient technologies that allow to establish a more carbon-neutral economy. Several attempts have been made to identify the major sources of environmental deterioration. Within the Stochastic Impacts by Regression on Population, Affluence, and Technology (STIRPAT) framework from 1960 to 2020, this study aims to check empirically the effect of fiscal decentralization (FD), green technological innovation (GI), trade openness (OPEN), population size (POP), per capita GDP (GDP), per capita GDP squared ($GDP^2$), institutional quality (INS), and energy consumption (EC) on carbon emissions ($CO_2$) in South Africa, as given its fast economic progress the country is facing problems with $CO_2$ emission. The recently developed novel dynamic autoregressive distributed lag (ARDL)-simulations framework has been used. The outcomes of the analysis indicate that (i) FD, GI, and INS improve environmental sustainability in both the short and long run; (ii) OPEN deteriorates environmental quality in the long run, although it is environmentally friendly in the short run; (iii) per capita GDP increases $CO_2$ emissions, whereas its square contributes to lower it, thus validating the presence of an environmental Kuznets curve (EKC) hypothesis; (iii) POP and EC contribute to environmental deterioration in both the short and long run; and (iv) FD, GI, OPEN, POP, GDP, $GDP^2$, INS, and EC Granger cause $CO_2$ in the medium, long, and short run, suggesting that these variables are important to influence environmental sustainability. In light of our empirical evidence, this paper suggests that the international teamwork necessary to lessen carbon emissions is immensely critical to solve the growing trans-boundary environmental decay and other associated spillover consequences. Moreover, it is important to explain responsibilities at different tiers of government to effectively meet the objectives of low $CO_2$ emissions and energy-saving fiscal expenditure functions.

**Keywords:** fiscal decentralization; trade openness; $CO_2$ emissions; green technological innovation; institutional quality; population size; energy consumption; EKC; cointegration; economic growth; South Africa

## 1. Introduction

Climate change has been a serious concern for many years, and it is now a serious threat to everyone around the globe [1]. High fossil fuel consumption, which is raising earth's temperature and changing the climate, is to blame for this environmental risk. Anthropogenic global warming brought on by fossil fuel combustion has resulted in severe hurricanes, extreme weather events, droughts, and significant snowstorms [2]. Such

persistent dangers are harmful to human life and the environment. In order to address the rising energy demand in emerging nations, the Paris Climate Agreement (PCA), which limits global temperatures to 2 °C, was implemented in 2015. Currently, governments throughout the world are attempting to implement environmental legislation to reduce $CO_2$ emissions.

There have been several solutions to this problem, but fiscal decentralization (FD) is the key to enhancing environmental quality. To enhance the delivery of public services, FD permits the efficient transfer of authority and responsibility from the federal government to the local government (regional, provincial, and municipal) [3]. Their effectiveness is increased by resource allocation and income generation at the local level. The provincial administration maintains tight relations with the populace, is quick to pick up on citizens' wants and preferences and gives prompt attention to them. Lower-level governance is seen to have positive interactions with the populace and its surroundings. In order to ensure justice, stability, and efficiency, they should discourage activities that cause pollution. Reviewing the available literature on FD and $CO_2$ emissions has prompted some scholars to identify large gaps that require filling.

First of all, despite the fact that FD plays a significant role in the literature already in existence, very few studies have been conducted that are adequate to explain the connection between FD and the environment, since they use a "race to the top" strategy to set high environmental standards and implement "beggar-thy-neighbor" policies that allow pollution to be released into nearby nations to improve the quality of the environment [4]. A few experts, nevertheless, rejected the notion of using a "race to the bottom" strategy to grow their enterprises at the expense of the environment by adopting lax ecological norms. Therefore, more research is needed since there is controversy regarding the relationship between FD and $CO_2$. In recent decades, subnational administrations instead of national authorities have been in charge of environmental policy. On the other hand, subnational governments typically fail to adhere to the rules set out by the central government, whereas subordinate administrative authorities comply. In the FD literature, the effect of FD on environmental quality has drawn a lot of attention.

Second, Green Technological Innovation (GI), which is utilized to reduce $CO_2$ emissions, is another issue that has to be addressed. It is well-known that effective innovation lowers costs and boosts a nation's revenue [5]. GI is a product and process innovation that uses waste recycling, pollution control, energy-saving technologies, and environmental management. This innovation efficiently promotes green development and satiates the criteria for controlling environmental protection. Additionally, it is one of the most significant factors contributing to improve climate change. Environmental technology innovation is seen as an effective tool for lowering $CO_2$ emissions [6]. GI is thought to increase productivity expansion and conserve resources. Cost reductions, enhanced efficiency, and better logistics are just a few of the immediate benefits of GI. Additional benefits include a better brand, improved health, and strong client connections. According to reports, GI motivates employees to save money and enhances financial results.

Third, economic growth is considered as being propelled by trade openness (OPEN), which is important for both emerging and industrialized nations. It facilitates the maximum utilization of a country's comparative advantage, leading to increased revenues and profits in nations with an enormous number of low-skilled jobs; it explicitly allows the import of foreign invested capital and industrial raw materials necessary to augment local industries in emerging regions; and it promotes openness to creative minds, as amply demonstrated by [1]. Faster development is the consequence, and increased entrepreneurial spirit is the result, of better market access and more intense competition. Additionally, access to foreign markets for goods is a catalyst for new investment, which raises productivity and boosts work opportunities and real earnings [7]. Most studies agree that globally engaged (more open) economies grow more quickly and, as a result, are typically more productive than their counterparts, which just focus on producing for home markets. Additionally, trade openness encourages resource utilization efficiency, which boosts productivity expansion.

This may eventually result in the buildup of more factors, knowledge spillovers, and the dissemination of technology [8]. The benefits of trade openness have drawn more attention than its environmental effects.

Fourth, in addition to the FD, GI, and OPEN, institutional quality (INS) is a crucial step in lowering environmental load. The improvement of environmental quality depends on INS. The domestic and national frameworks, in particular, shown that institutions are the key environmental country-quality determining factor [9]. Governance is measured by six factors: governmental integrity, regularity quality, law enforcement effectiveness, government efforts, anti-corruption efforts, democracy, and good governance. Ecological equilibrium is maintained via governance structures. An appropriate governance foundation enables the state to manage emissions and carbon concentrations via democratic climate policy [10]. Corruption has an impact on $CO_2$ emissions because it raises political expenses rather than reduces them. Additionally, INS enhances the environment by facilitating the spread of power and high-quality income. The impact of INS on pollution is both direct and indirect. The developed world has accomplished a lot, but the long-standing economic growth strategy, which is defined by high energy requirements, carbon dioxide emissions, and rising pollution, has caused tension for the ecosystem, making it difficult for economies to expand sustainably. According to several research works, various institutional settings have varying effects on the production of greenhouse gases [11]. As a result, the validity of earlier findings makes the current investigation necessary.

### 1.1. Overview of South African Economy

The current study attempts to fill the vacuum left by the constraints by employing FD, GI, OPEN, and INS as efficient strategies for reducing pollution in South Africa, since greenhouse gas emissions are dangerous for the country's sustainable development and have grown drastically globally. By examining the causal link between FD, GI, OPEN, INS, and $CO_2$ emissions, this work offers various empirical contributions to the literature. South Africa was chosen because the country has developed its economic system, which is extremely decentralized and resource-intensive. The following features also served as inspiration for choosing the country as a reference point. South Africa faces significant $CO_2$ emissions due to its fast development. To attain the Sustainable Development Goals (SDGs) and enhance environmental performance, the country is confronted with a number of serious issues. In 2018, its energy consumption and the rest of the BRICS nations constituted 40% of the global total. In terms of geo-economics, rapid economic growth, and participation in global economic development, South Africa and the rest of the BRICS nations are recently emerging developing economies that are significant. According to the World Bank, these nations contributed 23.31% of the global GDP in 2007. The BRICS economies' rapid economic growth resulted in $CO_2$ emissions per person of 13.985 billion tons, or 41.8% of global $CO_2$ emissions. In terms of consumption of coal and lignite, South Africa ranks first and sixth in the continent of Africa, respectively, and second globally in terms of $CO_2$ intensity. Concerning increases in $CO_2$ levels in South Africa and the rest of the BRICS nations (Brazil (1.15%), China (16%), India (5%), and Russia (6%)) in 2020 have led many to question whether (or not) these developing economies are up to the task of directing the globe toward a sustainable world. In reality, South Africa faces significant obstacles to lowering $CO_2$ emissions and achieving long-term economic growth. For the purpose of comparison, the total $CO_2$ emissions for South Africa and the rest of the BRICS nations in 2020 are depicted in million tons (MT) in Figure 1 below.

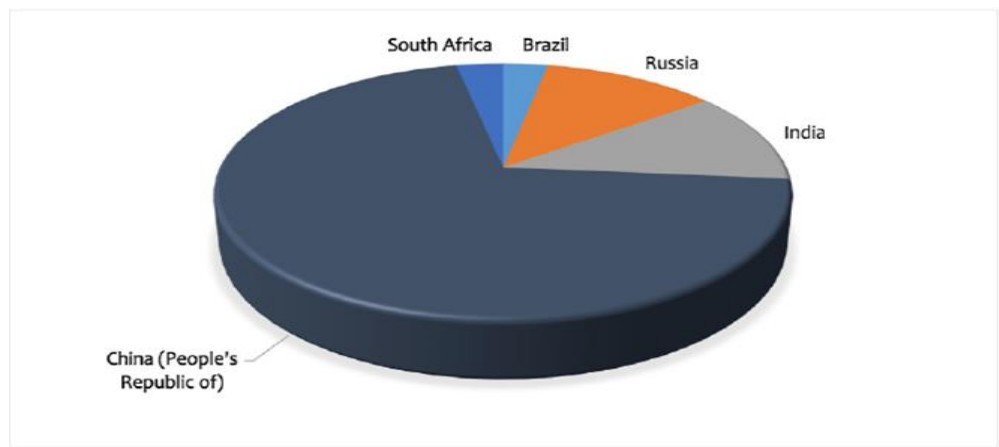

**Figure 1.** $CO_2$ emissions in BRICS in the year 2020. Source: Constructed from World Bank's World Development Indictors (2021).

We plot the trends in trade openness and $CO_2$ emissions in South Africa from 1960 to 2020, in Figure 2, to illustrate the relationship between trade openness and environmental quality (measured as a proxy by $CO_2$ emissions) in South Africa. The investigation separates the time-series data into two groups: "period before trade policy changes between 1960 and 1988" and "period after trade policy reforms with higher trade openness between 1989 and 2020", in order to compare and contrast these two variables. The patterns demonstrate a decline in trade openness during the 1960s, which was followed by an increase in openness throughout the 1970s. The time after the introduction of export promotion industrialization in South Africa, which began in 1972, is when trade openness is on the rise. The periods in the 1980s and 1990s saw a significant reduction in South Africa's trade openness, notably between 1990 and 1992. However, trade openness began to increase steadily from 1993 on, peaking at 72.9% in 2008. South Africa's trade openness experienced a dramatic decrease to 55.4% in 2009, as a result of the global crisis that began in 2008, before increasing once again in 2010. In the 10 years from 2011 to 2020, South Africa's trade openness stayed just over 60% [12].

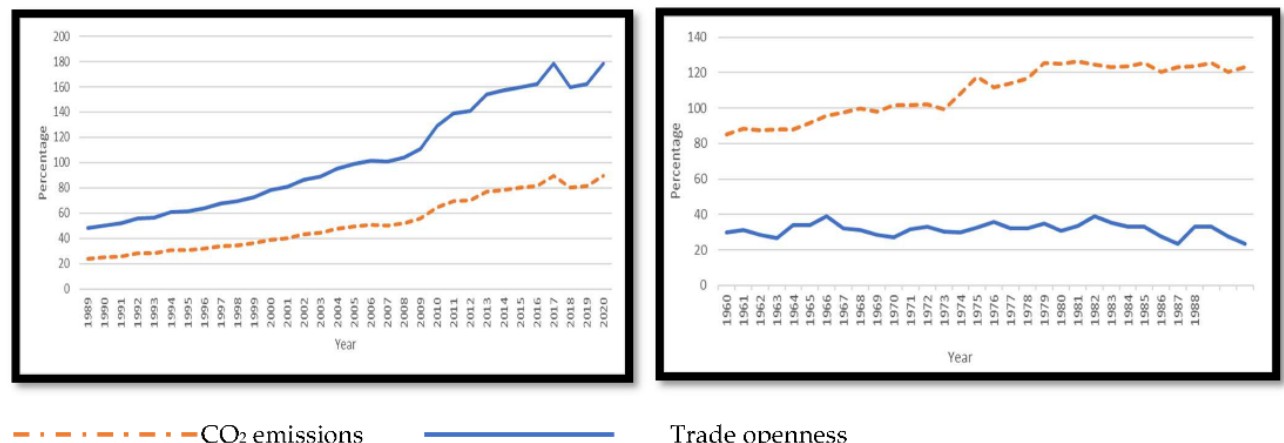

**Figure 2.** Trends in trade openness and $CO_2$ emissions in South Africa, 1960–2020. Source: World Bank's World Development Indictors (2021).



Figure 2 depicts the increased trend of $CO_2$ emissions between the years 1960 and 2020. South Africa is the world's 14th-highest emitter of GHGs, and its $CO_2$ emissions are mostly caused by a significant reliance on coal. Carbon emissions are progressively rising in South Africa [12]. A recently disclosed electricity plan draft, however, suggests a substantial move away from the fuel and toward gas and renewable energy sources. The plan calls for no new plants to be built after 2030 and the closure of four-fifths of the capacity by 2050, even though coal will continue to play a role for decades. South Africa has committed to achieving a peak in its emissions between 2020 and 2025, after which they will begin to decline for around 10 years.

### 1.2. Objectives of the Study

Given the foregoing arguments and justifications, the purpose of this research is to apply the novel dynamic ARDL simulations to the data from 1960 to 2020, in order to examine the effect of FD on $CO_2$ emissions in the context of GI, OPEN, INS, and GDP, within the STIRPAT framework for South Africa. We believe that this is the first analysis of the environment–FD nexus for South Africa that considers the joint effects of GI, OPEN, and INS. In addition, this work adds to the small amount of the literature that uses the novel econometric technique indicated above, which was introduced by [13]. Therefore, the empirical results of this study should be used to inform solid and trustworthy policy recommendations.

The remainder of this study is organized as follows: Section 2 reviews the literature, whereas Section 3 describes the theoretical analysis and empirical approach. The findings are discussed in Section 4, and Section 5 concludes the study by making suitable policy suggestions.

## 2. Literature Review

The current body of research goes into great detail about the variables affecting $CO_2$ emissions, which are thought to be significantly influenced by environmental innovation, carbon price, human capital, deforestation, energy consumption, financial development, population, output, and globalization. An introduction to the investigation of the interaction between income and environment is given by the renowned work of [14]. In order to analyze the causes of $CO_2$ emissions, researchers then incorporated energy use, deforestation, and global trade. Investigations on the climate change have increasingly begun to consider eco-innovation as a determinant of environmental protection, stating that energy-efficient techniques aid in the development of a more climate-resilient production structure and, as a result, lower $CO_2$ emissions. Therefore, the literature review part is broken down into four sub-sections, to discuss the empirical literature that has been done on the relationship between environmental quality and the factors that influence it.

### 2.1. Fiscal Decentralization and Environmental Quality Nexus

Authorities and intellectual communities have been debating the ecological effects of administrative and political decentralization in an increasing manner. Numerous empirical and theoretical studies on the effects of fiscal decentralization on the supply of public services such as environmental stewardship are available. In order to draw conclusions regarding the empirical effects of decentralization on environmental stewardship, this subcategory reviews this research. Numerous ideas in this subcategory of the literature contend that fiscal decentralization and levels of pollution in various regions of an economy go hand in hand. In [15]'s framework, decentralization increases inter-jurisdictional variance because it allows governments to regulate their harmful emissions. Li et al. [3] investigated the asymmetric fiscal decentralization effect on economic growth and environmental quality by using Pakistani data from 1984 to 2018. Their findings demonstrated that expenditure decentralization has asymmetric effects on economic growth and $CO_2$ emissions in the short and long run in Pakistan. Thus, positive, and negative fluctuations in expenditure decentralization affect economic growth and $CO_2$ emissions differently in Pakistan. The results of asymmetric ARDL suggested that negative shock of revenue decentralization

reduced economic growth and $CO_2$ emissions in the short and long run, while positive shock of revenue decentralization increased economic growth and $CO_2$ emissions.

The "climate" has grown to be one of the most significant and contentious global issues, and authorities are exploring different metrics of pollutant emissions. Consequently, several countries have been advocating fiscal decentralization to promote ecological integrity by giving provincial and subnational governments more financial freedom. So, using a new dynamic panel ARDL model from 1984 to 2017, Jain et al. [16] assessed the dynamic effect of fiscal decentralization on $CO_2$ in a set of nine Asian nations, and their empirical evidence suggested that fiscal decentralization has asymmetric impacts on $CO_2$ emissions because a shift in revenue and spending decentralization decreased $CO_2$ emissions in Asia. Additionally, a negative shift in expenditure decentralization has, over time, increased $CO_2$ emissions. Consequently, guidelines and rules for a clean environment might be changed. Similarly, by outlining how natural resource rent and fiscal decentralization influence $CO_2$ emissions, Tufail et al. [4] offered a fresh perspective. The authors used the panel data from seven highly fiscally decentralized Organization for Economic Cooperation and Development (OECD) nations from 1990 to 2018 to evaluate this aim. Their results showed that fiscal decentralization and rent from the natural resources benefit the environment by lowering $CO_2$ emissions. Additionally, when institutional quality improves, $CO_2$ emissions are decreased, while the gross domestic product and overall rent from natural resources rise. The survival of life on earth is seriously threatened by declining environmental quality.

Numerous nations have been trying to lessen their dependence on non-renewable energy sources by implementing new energy-efficient technologies that support the development of an industrial structure that is more sustainable. In this context, Cheng et al. [5], who investigated the effects of technological advancement and fiscal decentralization on carbon dioxide ($CO_2$) emissions in the context of China's GDP and globalization from 2005 Q1 to 2018 Q4, found that GDP, globalization, fiscal decentralization, and technological innovation all play significant roles in explaining China's $CO_2$ emissions. In terms of the consequences for policy, the authors proposed that China develop measures to reduce emission levels, by encouraging an energy-efficient system in order to address the worsening environmental quality. Meanwhile, to attain the goals of low $CO_2$ emissions and energy-saving functions of fiscal expenditures, it is also crucial to clearly define the duties at various levels of government. Although there is less of a scientific foundation for this claim, the question over whether fiscal decentralization may successfully reduce carbon dioxide ($CO_2$) emissions is receiving more and more attention. Khan et al. [17] examined the effect of fiscal decentralization on $CO_2$ emissions, using a balanced panel dataset of seven OECD nations between 1990 and 2018, to give empirical evidence in support of the hypothetical claim. According to their empirical findings, fiscal decentralization enhanced ecological integrity. Furthermore, increases in institutional quality and the growth of human capital reinforced the link between fiscal decentralization and ecological sustainability.

## 2.2. Green Technological Innovation (GI) and Environmental Quality Nexus

The empirical relationship between GI and $CO_2$ emissions has been examined in a number of research works [18–21]. By supporting businesses in lowering their reliance on energy, advancements in GI can lower $CO_2$ without affecting industrial productivity. Governments, businesses, and institutions of higher learning are all investing in technology to increase the efficiency and productivity of capital input [18,21]. For instance, Xin et al. [21] recently investigated this relationship between GI and $CO_2$ for the United States from 1990 Q1 to 2016 Q4 using the fully modified ordinary least squares approach. Their empirical research showed that GI during the growth period decreased $CO_2$ emissions by a positive partial sum, while $CO_2$ increased during the contraction phase due to the GI's negative partial sum. While the usage of renewable energy reduced $CO_2$ emissions, GDP growth increased them.

The asymmetric relationship between $CO_2$ emissions and GI in highly decentralized countries, on the other hand, was examined by [20]. Only $CO_2$ in the middle to higher quantiles was reduced, according to their findings. Additionally, GI had the least impact on quantiles with lower emissions and the most impact on quantiles with higher emissions. While both population and GDP growth increased $CO_2$ emissions, the effects were more evenly distributed, with lower emissions quantiles seeing the least increase and upper emissions quantiles experiencing the most increase. Similar to this, research by [22] for the BRICS countries showed that, in Brazil, Russia, India, and China, the emissions reduction effect of GI was only noticeable at upper emissions quantiles, whereas GI was positively related to $CO_2$ at lower emissions quantiles. Using the ARDL approach, Xiaosan et al. [23] demonstrated that green innovation, renewable energy production, and hydroelectricity production decreased $CO_2$ emissions in China, while GDP per capita grew. According to [24], the top 10 $CO_2$ emitters, GI fell as GDP increased $CO_2$ emissions. Furthermore, Ding et al. [19] found that GDP increased emissions, but eco-innovation declined in the G-7. According to [25]'s findings for fiscally decentralized countries, GDP surged while GI declined $CO_2$ emissions.

However, Zhang et al. [26] found that in China, the majority of innovation proxies significantly reduced $CO_2$ emissions, including the ratio of R&D inputs to GDP, the ratio of R&D personnel equivalent to the total population, the ratio of invention patents to GDP, the total turnover in the technical market, and the ratio of Internet users. Additionally, Sun et al. [27] proposed that eco-innovation reduced $CO_2$ emissions in the setting of a developed economy, i.e., the U.S. The environmental Kuznets curve (EKC) theory, which postulates a bell-shaped connection between income and environmental quality, was validated.

### 2.3. Trade Openness and Environmental Quality Nexus

The connection between trade and ecological sustainability has been the subject of several studies. However, across a variety of research settings and nations under examination, the results of these investigations are often ambiguous and conflicting. While some studies concluded that trade openness improves ecological integrity through a range of mechanisms, others claimed that trade openness worsens the state of the environment. Increased trade openness helps G-7 nations' ecological integrity, according to a scholarly investigation by [19], which uses cross-sectional autoregressive distributed lag (CS-ARDL) and augmented mean group (AMG) methods to collect data. Furthermore, ref. [28], using the common correlated effect mean group (CCEMG) and the mean group (MG) for the G-20 nations, concluded that trade openness lowers pollution problems. Similar to this, ref. [29], who utilize trade facilitation (TF) as a metric of trade openness for 48 Sub-Saharan African nations throughout the period from 2005 to 2014, note that TF is beneficial to the environment and supports ecological integrity in the region.

Khan et al. [30] provided evidence for the expanding importance of trade openness using the AMG and CCEMG techniques. In contrast, the experimental evidence by [17] indicated that trade openness erodes the quality of the environment in Pakistan. This credible research is indeed substantiated by [31], who observed that trade openness adds to significantly raise $CO_2$ emissions in the G-7. Ref. [32], who demonstrated that trade openness intensifies the quality of the environment in 66 emerging countries during the period of 1971–2017, confirmed the identical trends. Ref. [33] observed that trade openness rises $CO_2$ emissions for 88 countries during the period of 2000–2014, using both the difference and system generalized method of moments. Additionally, according to [34], opening up the OIC nations' markets to foreign goods has a negative impact and significantly worsens their environmental circumstances.

## 2.4. Institutional Quality and Environmental Quality Nexus

There is a lot of research on the variables that affect environmental performance, but little is known about the contribution of institutions to environmental sustainability. Regarding the environmental and social aspects of sustainable development, ref. [35] stressed the crucial role of national institutions in setting up the proper framework for neighborhood initiatives that support ecologically sustainable development. Ref. [34] looked at how institutional efficiency, along with other elements such as urbanization, FDI inflows, and trade, affected the ecological quality of 47 OIC (Organization of Islamic Cooperation) nations. They noticed that, with the exception of institutions, every other component had a positive correlation with the ecological quality. Ref. [36], who explained how the diversification of countries' energy mix and their institutional performance are related to ecological sustainability, while disaggregating energy sources, found that institutional efficiency improved ecological sustainability, economic expansion and the use of non-renewable energy sources harm the environment. "Better Environment, Better Tomorrow" is the catchphrase for the UN's Sustainable Development Goals (SDGs) and ecological improvements.

Ref. [7] examined the effects of regulatory quality, energy consumption per capita, and foreign direct investment on greenhouse gas emissions in panel Asian economies from 2001 to 2018. Regulatory quality and energy consumption per capita are used as explanatory variables, and foreign direct investment is used as an integrating variable with regulatory quality and energy consumption. Their results showed that energy consumption has a negative effect on GHG emissions, but regulatory quality actively encourages GHG emission reduction in Asia. Additionally, foreign direct investment plays an integrating function by considerably boosting clean energy consumption and energy consumption efficiency, when adequate foreign direct investment inflows are made [2], who used the 18th CPC National Congress's anti-corruption campaign in China as an exogenous influence to demonstrate how anti-corruption may greatly improve the quality of corporate environmental information disclosure, found that anti-corruption policies raised the cost of violations for businesses, promoted innovation among businesses, and eventually improved the quality of environmental information disclosure by businesses.

## 2.5. Summarizing Literature Gaps

Numerous significant areas of knowledge have been left unaddressed given the widespread prominence of earlier efforts; these inadequacies are prioritized in this investigation. The voids that have been found are as follows: Firstly, no research has been done in South Africa to look at the deep interaction between fiscal decentralization and ecological sustainability or to provide light on the precise mechanisms by which this linkage could operate. Secondly, there is no agreement in the prior studies regarding the link between fiscal decentralization and ecological sustainability, largely because very few investigations have used the right model for this connection, such as the endogenous growth model, and have instead used a variety of models to explore the relationship. Thirdly, several investigations have been carried out to identify the main factors contributing to environmental deterioration. Nevertheless, unlike international trade, economic output, and eco-innovation, studies frequently neglect a nation's political structure because of its indirect influence on decarbonization, which is hard to assess. In this research, we focus on South Africa and investigate how FD affects $CO_2$ emissions in the context of GI, OPEN, POP, GDP, $GDP^2$, INS, and EC. When creating strategies to promote environmental sustainability, it is crucial to comprehend the connection between FD and $CO_2$ emissions. By focusing on the impact of FD on $CO_2$ emissions, in the example of South Africa, this study closes a gap in the body of current knowledge. Fourthly, this work experimentally adds to the literature by examining the effects of FD on environmental quality using the sophisticated time-series estimation techniques. The outcomes can be used to establish environmental and FD policies in South Africa. Fifthly, the EKC hypothesis (Environmental Kuznets Curve Hypothesis) framework has been employed extensively in prior research on the FD–$CO_2$ nexus. The STIRPAT model (Stochastic Impacts by Regression on Population, Affluence,

and Technology) was not used in any of the experiments. As a result, by using the STIRPAT paradigm to examine the effect of FD on $CO_2$ emissions, our study adds to the body of previous research. Lastly, no study has been conducted to investigate the environmental effect of FD in the context of OPEN using the newly developed measurement proposed by Squalli and Wilson (2011), to consider a multidimensional index, with composite trade shares (CTS), to measure for trade openness. Trade share (TS), which exclusively reflects the domestic dimension, has been extensively used and predominates. Since this measure does not address the ambiguity surrounding trade openness, we, therefore, employ the Squalli and Wilson metric of trade openness, which is more informative, so it is expected to alter the strength of the relationships between trade openness and environmental quality in several aspects. Since CTS captures the multidimensional characteristics of trade openness and is, therefore, able to give a deeper understanding of the trade openness status of the country, it has a number of benefits over TS.

## 3. Theoretical Model and Empirical Methodology

### 3.1. Theoretical Model

Figure 3 presents the methodological roadmap of this research. Studies on the factors influencing carbon emissions are becoming much more common in an effort to identify any potential unique drivers of $CO_2$ emissions. The STIRPAT (Stochastic Impacts by Regression on Population, Affluence, and Technology) model is the one that is most frequently used to address this issue. The STIRPAT model, which is predicated to be linear and simple to read and estimate, aids in determining the degree to which a certain human activity impacts the environment. The model also allows for hypothesis testing and calculates the overall impact of each anthropogenic component on the ecosystem. The STIRPAT model has been used in earlier research to examine how different variables affect $CO_2$ emissions. In order to determine the connection between trade openness, industrialization, economic growth, energy consumption, and $CO_2$ emissions for Australia, ref. [37] used the STIRPAT model. Using the STIRPAT model, ref. [38] examined the long- and short-term effects of energy, economic, and sociological variables on environmental pollution for 28 European nations. Using the STIRPAT model, ref. [39] looked at the connection between population and carbon emissions. Using the expanded STIRPAT model, ref. [40] investigated the effects of economic development, technology, population, urbanization, industrialization, service level, trade, and energy structure on $CO_2$ emissions in China from 1980 to 2010. Ref. [41] used the STIRPAT model to examine how urbanization affects energy usage in China. Ref. [42] investigated the connection between population increase, economic expansion, and $CO_2$ emissions for emerging nations using the STIRPAT model. In contrast to these investigations, none of the available studies use the STIRPAT model to identify the effect of fiscal decentralization in promoting environmental quality. As a result, in light of the aforementioned investigations, the STIRPAT model representation is as follows:

$$I_t = aP_t^b A_t^c T_t^d \mu_t \tag{1}$$

where *I* is for overall environmental impact, *P* stands for population size, *A* stands for affluence, and *T* refers for technology or the impact per unit of socioeconomic activity. With the exception of population size and affluence, which have an influence on '*I*', '*T*' can be broken down into many factors depending on the environmental impact being studied [39]. In the equation above, "*a*" is a constant, "*b*", "*c*", and "*d*" are, respectively, the exponents of "*P*", "*A*", and "*T*", and "*μ*" is the error term in period "*t*" Although the technological component "*T*" may be broken down into a number of factors, the variables chosen to depend on the environmental effect being studied. More crucially, the STIRPAT model enables us to examine the applicability of the EKC hypothesis. To test if there is a non-linear link between GDP and $CO_2$ emissions, we have included a quadratic GDP factor to the model. The updated STIRPAT model is, therefore, provided as follows:

$$logCO_{2t} = a + blogFD_t + clogGI_t + dlogOPEN_t + elogPOP_t + flogGDP_t + glogGDP_t^2 + hlogINS_t + ilogEC_t + \varepsilon_t \tag{2}$$

where *FD* denotes fiscal decentralization, *GI* represents green technological innovation, *OPEN* captures trade openness, *POP* stands for population size, *GDP* means gross domestic product, $GDP^2$ represents the square of gross domestic product, *INS* stands for institutional quality, and ENE represents energy consumption.

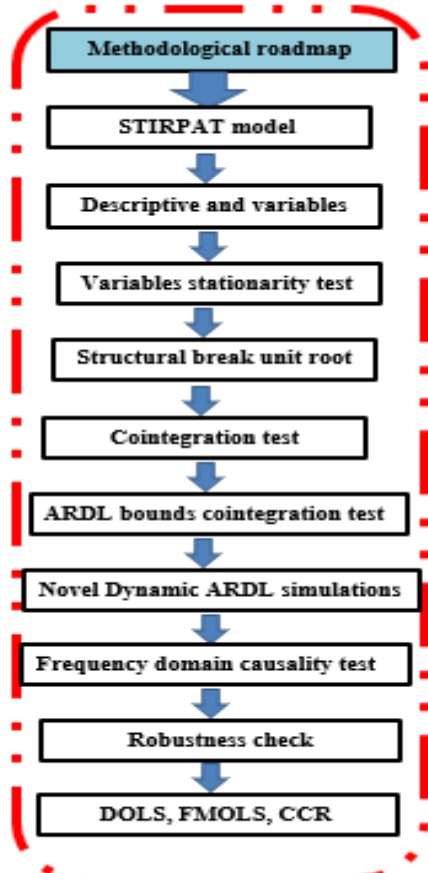

**Figure 3.** Methodological roadmap of the study.

In Equation (2), per capita *GDP* is utilized as a proxy variable for affluence, POP is used to represent population size, and technology factor 'T' is decomposed into several variables such as *FD*, *GI*, *OPEN*, *INS*, and *EC*. We have added $GDP^2$ to validate the existence of the EKC hypothesis for South Africa.

*3.2. Variables and Data Sources*

The dependent and independent variables used in the STIRPAT model are described in this section. The data used in this study are annual time-series data from 1960 to 2020.

Carbon Emissions ($CO_2$)

The dependent variable in the model is $CO_2$ emissions, which are measured in metric tons. The World Bank's World Development Indicators provided the data for $CO_2$ emissions from 1960 to 2020.

Fiscal Decentralization (FD)

The variable of interest in the model is FD, and it is calculated as the ratio of subnational spending to total federal expenditure (percentage). Information for FD from 1960 to 2020 was given by the International Monetary Fund (IMF). FD is a method for enhancing the provision of public services that aims to provide theories for dividing and organizing revenue and spending between central and local governments. A significant correlation exists between FD and $CO_2$ emissions. In light of the fact that governments may successfully adopt laws aimed at enhancing environmental quality by authorizing the lowest unit of the state, we, thus, hypothesize that as FD rises, the level of carbon emissions decreases.

The literature, however, is split on the FD coefficient's sign. Theoretically, governments can pass laws aimed at improving environmental quality by allowing the lower level of the state, which is the theoretical basis for the negative correlation between FD and $CO_2$ emissions. On the other hand, FD increases $CO_2$ emissions due to the free-rider effect among jurisdictions, which states that the possibility of environmental degradation increases with the amount of fiscal spending authority allocated to the lower units of the state [20]. The nature of the connection between FD and $CO_2$ emissions is, therefore, uncertain.

The following summarizes the explanations for the factors taken into account in the FD–$CO_2$ emissions equation:

Green Technological Innovation (GI)

GI is expressed as a percentage of all technologies, including those that are relevant to the environment. The information on green innovations and technology comes from OECD, 2021. GI, sometimes referred to as environment-related technologies (ERT), is one of the most successful methods to lower $CO_2$. Effective innovation is recognized to reduce expenses and increase a country's income. Recycling garbage, reducing pollution, utilizing energy-saving technology, and environmental management are all part of the GI product and process innovation. It encourages sustainable growth and satisfies the requirements for managing environmental protection. Additionally, it is considered to be a crucial element that results in minimal energy consumption, increases energy efficiency, and promotes environmental quality [19,21]. As such, it is one of the most critical components helping to ameliorate climate change. A more effective implementation of green technological breakthroughs is essential to support green economies and contribute to the reduction of rising $CO_2$ emissions. Since GI may help shift energy resources such as renewables from conventional to more efficient and sustainable sources, the study considers it as one of the drivers of $CO_2$ emissions and evaluates its influence, in line with [5,43]. Since numerous new green technologies are being produced as a consequence of increased green R&D activity, which might drastically affect the interactions between energy usage and production leading to $CO_2$ emissions mitigation, we hypothesize that as GI rises, the amount of carbon emissions decreases.

Trade Openness (OPEN)

Our study uses the [44] metric of trade openness, the composite trade share (CTS), which is more informative, to give a deeper understanding of the trade openness status of the country, as it has a number of benefits over the conventional trade share (TS). Data for OPEN are sourced from the World Bank's World Development Indicators from 1960 to 2020. Economic growth is considered as being propelled by OPEN, which is important for both emerging and industrialized nations. It facilitates the maximum utilization of a country's comparative advantage, leading to increased revenues and profits in nations with an enormous number of low-skilled jobs. However, the theory linking trade to the environment has generated controversy, and its empirical verification has been conflicting and mixed. Thus, we hypothesis that as OPEN increases, the level of carbon emissions could either reduce or increase. Meanwhile, under the pollution haven hypothesis (PHH), an increase in OPEN is expected to exacerbate the level of carbon emissions in developing nations such as South Africa. Therefore, following [45,46], the study uses OPEN to capture its environmental effect, since openness to international goods markets propagates more economic activities and, thus, increases $CO_2$ emissions. The novelty of *CTS* as a measure of OPEN in this work is that it mirrors trade-outcome reality because it contains two dimensions of a country's ties with the rest of the world. The *CTS* is presented, thus, as follows:

$$CTS = \frac{(X+M)_i}{\frac{1}{n}\sum_{j=1}^{n}(X+M)_j} \frac{(X+M)_i}{GDP_i} \tag{3}$$

where *i* denotes South Africa, *j* reflects its trading partners, *X* represents exports, and *M* denotes imports. In Equation (3), while the first segment captures world trade share, the second portion accounts for South Africa's trade share.

Population Size (POP)

POP is calculated as the nation's total yearly population (million). This metric is regarded as one of the most significant variables influencing variations in $CO_2$ emissions. Data for POP utilized in the study came from the World Bank's World Development Indicators for the period of 1960 to 2020. According to our hypothesis, rising POP will result in higher energy consumption and higher $CO_2$ emissions.

Per Capita GDP (GDP)

Real per capita GDP using constant US dollars, 2015, and is utilized as an independent variable. The World Bank's World Development Indicators contain data for per capita GDP from 1960 through 2020. To account for the impact of income in the model, per capita GDP is utilized [47]. As a result, we surmise that South Africa's per capita GDP and carbon emissions are positively correlated.

Per Capita GDP Squared $\left(GDP^2\right)$

To assess if there is a linear or nonlinear relationship between per capita GDP and carbon emissions, per capita GDP squared is employed. The EKC hypothesis implies an inverted U-shaped link between economic growth (per capita GDP) and carbon emissions, which predicts a negative association between the squared term of per capita GDP and carbon emissions. As a result, rising economic growth initially drives up carbon emissions. After a turning point, additional gains in economic growth thereafter result in a decrease in carbon emissions. We, thus, postulate that the squared per capita GDP term and carbon emissions are negatively correlated.

Institutional Quality (INS)

INS is a crucial element in lowering environmental load. The improvement of environmental quality depends on INS [48]. The domestic and national frameworks, in particular, show that institutions are the key environmental country-quality determining factor. Therefore, we hypothesize that a negative relationship exists between INS and carbon emissions.

Energy Consumption (EC)

The million tons of oil equivalent is used to measure EC. The World Bank's World Development Indicators contain information about EC from 1960 through 2020. Since the energy sector is responsible for 75% of worldwide GHC emissions, EC is chosen in this study to analyze the impact of energy consumption on $CO_2$ emissions [49–54]. We, thus, postulate that there is a positive correlation between EC and carbon emissions.

Table 1 lists the source of data, explanations, and measurements of variables utilized in Equation (2).

### 3.3. Narayan and Popp's Structural Break Unit Root Test

It is crucial to perform a stationarity test on the variables under consideration, in order to determine their order of integration, before putting the unique dynamic ARDL simulations model into practice. To verify the asymptotic behavior and order of integration of all variables under consideration, this study uses the unit root tests Dickey–Fuller GLS (DF-GLS), Phillips–Perron (PP), Augmented Dickey–Fuller (ADF), and Kwiatkowski–Phillips–Schmidt–Shin (KPSS). The problems caused by erroneous regressions are addressed by this procedure. Since empirical data indicate that structural breaks are relatively persistent and numerous macroeconomic variables, such as $CO_2$ emissions and trade openness, are likely to be affected, the second step employs Narayan and Popp's structural break unit root test.

**Table 1.** Description of variables and data sources.

| Variable | Description | Units | Sources |
|---|---|---|---|
| $CO_2$ | Carbon emissions | Metric tons | WDI |
| FD | Fiscal decentralization as measured by the subnational expenditure as ratio of total federal expenditure | Percentage | IMF |
| GDP | Gross domestic product | Constant USD, 2015, | WDI |
| GI | Green technological innovation | % of all environment-related technologies. | OECD |
| POP | Population size as measured by the number of individuals per square kilometer of the land area. | (million) | WDI |
| INS | Institutional quality as measured by the Principal Component Analysis (PCA) of six indictors, namely governmental integrity, regularity quality, law enforcement effectiveness, government efforts, anti-corruption efforts, democracy, and good governance. | PCA index | World Worldwide Governance Indicators |
| EC | Energy consumption, million tons of oil equivalent. | Metric tons | BP Statistical Review of World Energy |
| OPEN | Trade openness computed as composite trade share, introduced by Squalli and Wilson (2011), capturing trade effect. | % of GDP | WDI |

WDI: World Development Indicators; IMF: International Monetary Fund; OECD: Organization for Economic Cooperation & Development.

### 3.4. Cointegration Techniques

After checking for stationarity, ref. [55]'s test is used to look into the cointegration connection between the variables. By merging numerous distinct test findings, such as those of [55–57], this recently revised technique to cointegration provides a more accurate result. The following are Fisher's equations for the Bayer–Hanck method:

$$EG - JOH = -2\big[In(P_{EG}) + In(P_{JOH})\big] \tag{4}$$

$$EG - JOH - BO - BDM = -2\big[In(P_{EG}) + In(P_{JOH}) + In(P_{BO}) + In(P_{BDM})\big] \tag{5}$$

The probability values for each of the four cointegration tests mentioned above are $P_{BDM}$, $P_{BO}$, $P_{JOH}$, and $P_{EG}$. The Fisher statistics' structure governs the cointegration of the underlying variables. The long-term equilibrium relationship between $CO_2$ emissions and their putative causes, namely, FD, GI, OPEN, POP, GDP, $GDP^2$, INS, and EC, is then investigated by using the Maki test of cointegration, developed by [58], while accounting for different structural breaks. When compared to other cointegration tests that contain structural fractures, the Maki cointegration test, according to [59], offers superior size and power properties (e.g., [60]). Furthermore, when several structural breaks occur, focusing the research on just one structural break might lead to false conclusions.

### 3.5. ARDL Bounds Testing Approach

The bounds test is used in this study to look at the relationship between the variables under consideration over the long term. Following [61], the ARDL bounds testing method is stated as follows:

$$\Delta InCO_{2t} = \gamma_0 + \sum_{i=1}^{n} \gamma_{1i}\Delta InCO_{2t-i} + \sum_{i=0}^{n} \gamma_{2i}\Delta InFD_{t-i} + \sum_{i=0}^{n} \gamma_{3i}\Delta InGI_{t-i} +$$
$$\sum_{i=0}^{n} \gamma_{4i}\Delta OPEN_{t-i} + \sum_{i=0}^{n} \gamma_{5i}\Delta POP_{t-i} + \sum_{i=0}^{n} \gamma_{6i}\Delta InGDP_{t-i} +$$
$$\sum_{i=0}^{n} \gamma_{7i}\Delta InGDP_{t-i}^2 + \sum_{i=0}^{n} \gamma_{8i}\Delta InINS_{t-i} + \sum_{i=0}^{n} \gamma_{9i}\Delta InEC_{t-i} + \theta_1 InCO_{2t-i} + \quad (6)$$
$$\theta_2 InFD_{t-i} + \theta_3 InGI_{t-i} + \theta_4 InOPEN_{t-i} + \theta_5 InPOP_{t-i} + \theta_6 InGDP_{t-i} +$$
$$\theta_7 InGDP_{t-i}^2 + \theta_8 InINS_{t-i} + \theta_9 InEC_{t-i} + \varepsilon_t$$

where $\Delta$ represents the first difference of $InCO_2$, $InFD$, $InGI$, $InOPEN$, $InPOP$, $InGDP$, $InGDP^2$, $InINS$, and $InEC$. Meanwhile, $t - i$ denotes the optimal lags selected by Schwarz's Bayesian Information Criterion (SBIC), and $\gamma$ and $\theta$ are the estimated coefficients for short run and long run, respectively. The ARDL model for the long and short run will be approximated if variables are cointegrated. The null hypothesis, which tests for long-run relationship, is ($H_0 : \theta_1 = \theta_2 = \theta_3 = \theta_4 = \theta_5 = \theta_6 = \theta_7 = \theta_8 = \theta_9 = 0$), against the alternative hypothesis of ($H_1 : \theta_1 \neq \theta_2 \neq \theta_3 \neq \theta_4 \neq \theta_5 \neq \theta_6 \neq \theta_7 \neq \theta_8 \neq \theta_9 \neq 0$).

Whether the null hypothesis is accepted or rejected relies on the estimated F-statistic value. The null hypothesis is rejected, and cointegration or a long-term link between the variables is inferred, if the estimated F-statistic value exceeds the upper threshold. If the estimated F-statistic value is smaller than the lower bound, cointegration does not occur. Additionally, the bounds test is unconvincing if the estimated F-statistic value is between the lower and higher boundaries. If there is a long-term link between the variables, the following is the long-term ARDL model that has to be estimated:

$$\Delta InCO_{2t} = \beta_0 + \sum_{i=1}^{q} \omega_1 InCO_{2t-i} + \sum_{i=1}^{q} \omega_2 InFD_{t-i} + \sum_{i=1}^{q} \omega_3 InGI_{t-i}$$
$$+ \sum_{i=1}^{q} \omega_4 InOPEN_{t-i} + \sum_{i=1}^{q} \omega_5 InPOP_{t-i} + \sum_{i=1}^{q} \omega_6 InGDP_{t-i} \quad (7)$$
$$+ \sum_{i=1}^{q} \omega_7 GDP_{t-i}^2 + \sum_{i=1}^{q} \omega_8 InINS_{t-i} + \sum_{i=1}^{q} \omega_8 InEC_{t-i} + \varepsilon_t$$

$\omega$ denotes the long-run variance of variables in Equation (7). In choosing the correct lags, the paper uses the SBIC. For the short-run ARDL model, the error-correction model used is as follows:

$$\Delta InCO_{2t} = \beta_0 + \sum_{i=1}^{q} \pi_1 \Delta InCO_{2t-i} + \sum_{i=1}^{q} \pi_2 \Delta InFD_{t-i} + \sum_{i=1}^{q} \pi_3 \Delta InGI_{t-i} + \sum_{i=1}^{q} \pi_4 \Delta InOPEN_{t-i}$$
$$+ \sum_{i=1}^{q} \pi_5 \Delta InPOP_{t-i} + \sum_{i=1}^{q} \pi_6 In\Delta GDP_{t-i} + \sum_{i=1}^{q} \pi_7 \Delta InGDP_{t-i}^2 + \sum_{i=1}^{q} \pi_8 \Delta InINS_{t-i} \quad (8)$$
$$+ \sum_{i=1}^{q} \pi_9 \Delta InEC_{t-i} + ECT_{t-i} + \varepsilon_t$$

In Equation (8), $\pi$ represents the variables' short-run variability, while ECT stands for the error-correction term, which describes the disequilibrium's rate of adjustment. The range of the calculated ECT coefficient is from $-1$ to $0$. The diagnostic tests for model stability are also used in this work. The Ramsey RESET test is used to make sure that the model is correctly stated, and the Jarque–Bera test is used to determine if the estimated residuals are normally distributed. The Breusch-Pagan-Godfrey test and the ARCH test are both used to test for heteroscedasticity. This work uses the cumulative sum of recursive residuals (CUSUM) and the cumulative sum of squares of recursive residuals to test for structural stability (CUSUMSQ).

### 3.6. Dynamic Autoregressive Distributed Lag Simulations

The simple ARDL approach put forth by [61] and other cointegration frameworks, which can only estimate and explore the short- and long-run relationships between the variables, have been widely used in previous studies that looked into the relationship

between FD–$CO_2$ emissions. Ref. [13] have created the unique dynamic ARDL simulations model to address the flaws that are inherent in the basic ARDL model. This model can effectively and efficiently overcome the current issues surrounding the result interpretations inherent in the simple ARDL method. This newly developed framework is capable of simulating and plotting to predict graphs of (positive and negative) changes in the variables automatically and estimate the relationships for the short run and long run. The major advantage of this framework is that it can predict, simulate, and immediately plot probabilistic change forecasts on the dependent variable in one explanatory variable, while holding other regressors constant. In this study, based on the multivariate normal distribution for the parameter vector, the dynamic ARDL error correction algorithm uses 1000 simulations. We employ the graphs to examine the actual change of an explanatory variable as well as its influence on the dependent variable. The novel dynamic ARDL simulations model is presented as follows:

$$
\begin{aligned}
\Delta InCO_{2t} = \alpha_0 \quad &+ v_0 InCO_{2t-1} + \varphi_1 \Delta FD_t + \rho_1 FD_{t-1} + \varphi_2 \Delta GI_t + \rho_2 GI_{t-1} + \varphi_3 \Delta OPEN_t + \rho_3 OPEN_{t-1} \\
&+ \varphi_4 \Delta POP_t + \rho_4 POP_{t-1} + \varphi_5 \Delta GDP_t + \rho_5 GDP_{t-1} + \varphi_6 \Delta GDP_t^2 + \rho_6 GDP_{t-1}^2 + \varphi_7 \Delta INS_t \\
&+ \rho_7 INS_{t-1} + \varphi_7 \Delta EC_t + \rho_7 EC_{t-1} + \delta ECT_{t-1} + \varepsilon_t
\end{aligned}
\tag{9}
$$

### 3.7. Frequency Domain Causality Test

In order to investigate the causal linkages among the variables under investigation, this research uses the frequency domain causality (FDC) technique, a reliable testing procedure recommended by [62]. FDC makes it possible to predict the response variable at a given time frequency, which is virtually impossible with the traditional Granger causality approach. It also makes it possible to capture permanent causality for medium-, short-, and long-term relationships among the variables being studied. In this study, the robustness of the test is also checked with the help of this testing strategy.

### 3.8. Robustness Check

Three estimation frameworks, the fully modified ordinary least squares (FMOLS), canonical-correlation regression (CCR), and dynamic ordinary least squares (DOLS) tests, developed by [63–66], respectively, are used to assess the robustness of the estimates generated by the novel dynamic ARDL simulations model. Ref. [64] developed the semi-parametric approach FMOLS to get over the correlation issue, highlighting the test's asymptotically impartiality and effectiveness. In a model where the order of the integration of time series variables is I(1), cointegration vectors are investigated using CCR, a technique similar to FMOLS described by [63]. The fundamental distinction between FMOLS and CCR estimation methods is that FMOLS focusses on both data and parameter modification, while CCR places emphasis on data conversion [67]. To counteract small sample bias and simultaneity problem, the DOLS model adds leads and lags. By adequately handling the nuisance features, both DOLS and FMOLS estimation methods address the issues of serial correlation and endogeneity [68]. The spectral BC causality test, designed and proposed by Breitung and Candelon (2006), is also used in this work to capture the causative impacts of FD, GI, OPEN, POP, GDP, $GDP^2$, INS, and EC on $CO_2$ emissions in South Africa. Refs. [56,57] were the first to design the spectral BC test.

## 4. Empirical Results and Their Discussion

### 4.1. Summary Statistics

The summary statistics of the variables used in this work are analyzed and scrutinized before discussing the results. Table 2 reports the overview of statistics, showing that the $CO_2$ emissions average value is 0.264. Per capita GDP squared ($GDP^2$) has an average mean of 60.316 greater than the other variables. This is followed by population size (POP), which has 10.105. In addition to characterising the summary statistics, Table 2 uses kurtosis to represent the peak, while the Jarque–Bera test statistics are used to check for normality of our data series. The table shows that per capita GDP, fiscal decentralization (FD), trade

openness (OPEN), energy consumption (EC), population size (POP), institutional quality (INS), and green technological innovation (GI) show a positive trend, while per capita GDP squared has a negative trend. The variance in $GDP^2$ is the highest of all the variables, showing the high level of volatility in this variable. The variance in $CO_2$ emissions is less relative to $GDP^2$, showing that $CO_2$ emissions are far more stable. Moreover, the variations in OPEN, GDP, and GI are quite greater. In addition, the Jarque–Bera statistics shows that our data series are normally distributed.

**Table 2.** Descriptive statistics.

| Variables | Mean | Median | Maximum | Minimum | Std. Dev | Skewness | Kurtosis | J–B Stat | Probability |
|---|---|---|---|---|---|---|---|---|---|
| $CO_2$ | 0.264 | 0.238 | 0.477 | 0.084 | 0.120 | 0.217 | 1.652 | 4.682 | 0.196 |
| GDP | 7.706 | 7.959 | 8.984 | 6.073 | 0.843 | −0.511 | 2.156 | 4.102 | 0.129 |
| $GDP^2$ | 60.316 | 63.754 | 80.717 | 36.880 | 12.663 | −0.387 | 2.082 | 3.422 | 0.181 |
| FD | 5.716 | 5.014 | 10.625 | 2.814 | 0.150 | 0.714 | 2.615 | 3.014 | 0.148 |
| OPEN | 6.060 | 6.512 | 7.665 | 2.745 | 1.329 | 0.636 | 2.077 | 5.757 | 0.156 |
| EC | 4.220 | 4.422 | 4.840 | 3.177 | 0.527 | −0.558 | 1.921 | 5.621 | 0.160 |
| POP | 10.105 | 13.286 | 14.659 | 11.913 | 0.738 | 0.056 | 2.463 | 0.702 | 0.704 |
| INS | 3.513 | 3.580 | 3.813 | 3.258 | 0.161 | −0.215 | 1.697 | 4.474 | 0.107 |
| GI | 9.360 | 9.255 | 10.545 | 8.210 | 0.766 | 0.082 | 1.634 | 4.499 | 0.105 |

Source: Authors' calculations.

### 4.2. Order of Integration of the Respective Variables

All variables that are not stationary at level become stationary at I (1) after first differencing, according to Table 3's findings from the DF-GLS, PP, ADF, and KPSS tests. This suggests that none of the series under consideration are I (2) and that all are either I (1) or I (0). The conventional unit root tests described below do not take structural breaks into consideration. As a result, the testing technique used in this work would take into consideration two structural breaks in the variables. Narayan and Popp's unit root test with two structural breaks is, therefore, employed in the study, and the findings are also shown in the right-hand panel of Table 3. The empirical findings indicate that, in the presence of two structural breaks, the variables are integrated. As a result, the dynamic ARDL limits testing technique may be used to all data series that are integrated to order one.

**Table 3.** Unit root analysis.

| Variable | Dickey–Fuller GLS | Phillips–Perron | Augmented Dickey–Fuller | Kwiatkowski–Phillips–Schmidt–Shin | Narayan and Popp's (2010) Unit Root Test | | | |
|---|---|---|---|---|---|---|---|---|
| | (DF-GLS) | (PP) | (ADF) | (KPSS) | Model 1 | | Model 2 | |
| Level | Test-Statistics value | | | | Break-Year | ADF-stat | Break-Year | ADF-stat |
| $lnCO_2$ | −0.570 | −0.464 | −1.152 | 0.966 | 1982:1985 | −3.132 | 1987:1994 | −8.160 *** |
| lnFD | −0.166 | −0.242 | −1.136 | 0.331 *** | 1985:2007 | −1.504 | 2007:2013 | −7.204 *** |
| lnGDP | −0.116 ** | −0.079 | −1.308 | 0.833 *** | 1979:1988 | −2.914 | 1982:1990 | −7.601 *** |
| $lnGDP^2$ | −0.112 * | −0.076 | −1.268 | 0.848 *** | 1979:1990 | −1.939 | 1982:1994 | −6.791 *** |
| lnOPEN | −0.072 | −0.082 | −1.335 | 1.080 * | 1996:2001 | −3.053 | 2003:2009 | −7.318 *** |
| lnEC | −0.011 | −0.014 | −0.366 | 1.300 *** | 1982:1989 | −4.372 ** | 1985:1991 | −8.521 *** |
| lnPOP | −0.032 * | −0.001 | −0.012 | 0.640 | 2001:2006 | −2.021 | 2004:2010 | −8.362 *** |
| lnGI | −0.254 *** | −0.284 *** | −2.999 | 0.255 *** | 1995:2000 | −4.318 | 2008:2011 | −7.821 *** |
| lnINS | −0.046 | −0.071 * | −1.718 | 1.060 ** | 1972:1985 | −3.815 | 1982:1991 | −7.521 *** |
| | First difference | | | | Critical value (1%, 5%, and 10%) | | | |
| $\Delta lnCO_2$ | −0.995 *** | −0.996 *** | −7.176 *** | 0.705 *** | 1999:2005 | −4.801 ** | 1980:1991 | −5.832 *** |
| ΔlnFD | −0.731 *** | −0.514 *** | −5.841 *** | 0.504 *** | 1974:2003 | −5.815 *** | 2001:2008 | −8.605 *** |
| ΔlnGDP | −0.695 *** | −0.707 *** | −5.319 *** | 0.585 *** | 1983:1997 | −5.831 *** | 1985:1995 | −6.831 *** |
| $\Delta lnGDP^2$ | −0.694 *** | −0.707 *** | −5.316 *** | 0.589 *** | 1991:2000 | −8.531 *** | 1987:1996 | −5.893 *** |
| ΔlnOPEN | −0.935 *** | −0.938 *** | −6.699 *** | 0.626 *** | 1996:2004 | −6.842 ** | 2001:2007 | −8.942 *** |
| ΔlnEC | −1.105 *** | −1.121 *** | −8.142 *** | 0.586 *** | 1985:1993 | −5.921 *** | 1989:1997 | −7.942 *** |
| ΔlnPOP | −0.207 ** | −0.209 ** | −6.443*** | 0.609 *** | 2005:2008 | −6.831 *** | 2001:2008 | −6.973 *** |
| ΔlnGI | −1.023 *** | −1.034 *** | −7.473 *** | 0.424 *** | 1999:2003 | −4.841 ** | 2006:2010 | −5.983 *** |
| ΔlnINS | −0.799 *** | −0.801 *** | −5.878 *** | 0.431 *** | 1975:1990 | −7.742 *** | 1988:1992 | −7.892 *** |

Source: Authors' calculations. Note: *, **, and *** denote statistical significance at 10%, 5%, and 1% levels, respectively. MacKinnon's one-sided *p*-values. Lag length based on SIC and AIC. Probability based on Kwiatkowski–Phillips–Schmidt–Shin (1992). The critical values for Narayan and Popp's unit root test with two breaks follow [1,69]. All the variables are trended.

### 4.3. Lag Length Selection Results

The results of several test criteria for lags selection are presented in Table 4. In the empirical literature, it is noted that the most-often-used methods for choosing acceptable lags are HQ, AIC, and SIC. Lag selection in this study involves the usage of SIC. This tool suggests that lag one is appropriate for our model. This is such that, unlike other methods, SIC yields the lowest value at lag one.

**Table 4.** Lag-length criteria.

| Lag | LogL | LR | FPE | AIC | SC | HQ |
|---|---|---|---|---|---|---|
| 0 | 152.453 | NA | $3.2 \times 10^{-12}$ | −6.594 | −6.331 | −6.493 |
| 1 | 517.095 | 857.28 | $1.5 \times 10^{-18}$ | −21.195 | −19.094 * | −20.390 * |
| 2 | 531.093 | 108 | $1.4 \times 10^{-18}$ | −21.388 | −17.448 | −19.877 |
| 3 | 699.755 | 117.32 | $1.2 \times 10^{-18}$ * | −21.759 | −15.981 | −19.544 |
| 4 | 694.113 | 128.72 * | $1.3 \times 10^{-18}$ | −22.350 * | −14.733 | −19.430 |

Source: Authors' calculations. Note: * indicates lag order selected by the criterion.

### 4.4. Cointegration Test Results

The results of the cointegration test using the surface-response regression proposed by [70] are shown in Table 5. We reject the null hypothesis because the F- and t-statistics are higher than the upper-bound critical values at different degrees of significance. Therefore, cointegration between the variables under discussion is supported by our empirical results.

**Table 5.** ARDL bounds test analysis.

| Test Statistics | Value | K | $H_0$ | $H_1$ | | |
|---|---|---|---|---|---|---|
| F-statistics | 15.052 | 8 | No level relationship | Relationship exists | | |
| t-statistics | −8.752 | | | | | |
| Kripfganz and Schneider (2018) critical values and approximate *p*-values | | | | | | |
| Significance | F-statistics | | t-statistics | | *p*-value F | |
| | 1 (0) | 1 (1) | 1 (0) | 1 (1) | 1 (0) | 1 (1) |
| 10% | 2.94 | 3.04 | −2.57 | −4.04 | 0.000 *** | 0.000 *** |
| 5% | 2.37 | 3.43 | −2.86 | −4.38 | *p*-value t | |
| 1% | 3.04 | 4.02 | −3.43 | −4.99 | 0.000 *** | 0.002 ** |

Note: **, and *** respectively represent statistical significance at 5%, and 1% levels, respectively. The respective significance levels suggest the rejection of the null hypothesis of no cointegration. The optimal lag length on each variable is chosen by the Schwarz's Bayesian information criterion (SBIC).

Since the variables are integrated in the same order, we also employed the Bayer–Hanck and Maki cointegration tests to assess robustness. The variables are tested for cointegration using the Bayer–Hanck and Maki cointegration tests. The surface-response regression proposed by [70] earlier confirmed the occurrence of cointegration, which Table 6 validates. There is a long-term link between the factors under consideration. Furthermore, the Maki cointegration method generated significant test statistics for both the level and regime shifts, supporting our claim that the variables under study have a stable long-run relationship. The variables are cointegrated in the presence of structural breaks, according to the Maki cointegration test findings, accounting for structural breaks.

**Table 6.** Results of cointegration tests.

| Bayer–Hanck Cointegration (Without Structural Breaks) | | |
|---|---|---|
| **Test** | **Statistic** | **Critical value at 5%** |
| Engle–Granger–Johansen (EG–J) | 51.216 *** | 11.810 |
| Engle–Granger–Banerjee–Boswijk (EG–J–Ba–Bo) | 55.159 *** | 20.814 |
| **Maki Cointegration (With Structural Breaks)** | | |
| Model | Test Statistics | Structural Breaks |
| Level Shifts with Trend | −7.153 * | 1981–1994–2008 |
| Regime Shifts | −12.501 *** | 1980–1994–2008 |
| Regime Shifts and Trend | −10.160 *** | 1984–1993–2009 |

Note: *** and * denote 1% and 10% significance levels, respectively.

### 4.5. Diagnostic Statistics Tests

The study consequently employs many diagnostic statistical tests, and their empirical findings are shown in Table 7, in order to guarantee the consistency and reliability of our selected model. Given that the model in use passed all diagnostic tests, the empirical findings imply that it is well-fitted. The Breusch–Godfrey LM test demonstrates that the model is not affected by serial correlation or autocorrelation issues. Evidence obtained using the Ramsey RESET test demonstrates that the model is not misspecified. Both the Breusch–Pagan–Godfrey test and the ARCH test are used to determine if the model exhibits heteroscedasticity. According to the empirical results, heteroscedasticity is mild and not a concern. Finally, the results of the Jarque–Bera test demonstrate that the model's residuals are normally distributed.

**Table 7.** Diagnostic statistics tests.

| Diagnostic Statistics Tests | $X^2$ (*p* Values) | Results |
|---|---|---|
| Breusch–Godfrey LM test | 0.3812 | No problem of serial correlations |
| Breusch–Pagan–Godfrey test | 0.2610 | No problem of heteroscedasticity |
| ARCH test | 0.6837 | No problem of heteroscedasticity |
| Ramsey RESET test | 0.5183 | Model is specified correctly |
| Jarque–Bera Test | 0.2715 | Estimated residuals are normal |

Source: Authors' calculations.

### 4.6. Dynamic ARDL Simulations Model Results

Results from the dynamic ARDL simulation model are shown in Table 8. Our research demonstrates that $CO_2$ emissions are influenced adversely and favorably by per capita GDP (InGDP) and per capita GDP squared (InGDP$^2$), respectively. Environmental quality is worsened by InGDP, which represents economic expansion, whereas its square has a calming impact on the environment. As a result, empirical data reveal that the EKC hypothesis is true in the instance of South Africa, where real income increases up to a certain point but $CO_2$ emissions begin to fall. Environmental quality in South Africa declines throughout the early stages of economic expansion but begins to rise once the country reaches its optimal level. This is in favor of the inverted U-shaped link between environmental quality and economic growth. The findings are relevant to South Africa and are related to the country's structural transformation and technological advancement. As people's incomes rise, their awareness of the environment improves as well. As a result, environmental restrictions are enforced to employ energy-efficient technology to reduce pollution. These findings are consistent with [46], which revealed the presence of the EKC hypothesis for the Southern African Development Community (SADC) from 1960 to 2014. Similar findings supporting the EKC concept were made for the Middle East and North Africa (MENA) nations by [71]. The EKC hypothesis, which proposes an inverted U-shaped link between air pollution and economic growth in six different regions, including Latin

America and the Caribbean, East Asia and the Pacific, Europe and Central Asia, South Asia, the Middle East and North Africa, and Sub-Saharan Africa, was also supported by [72]'s investigation of the association between air pollution and economic growth. According to [1,8,46,52,53,69,73,74], South Africa fits within the scope of the EKC theory. Our findings also confirm those of [75] for eight OECD countries, ref. [76] for China, [77] for China, [78] for 155 countries across four income categories, and [79] for six economies in South Asia. The results, however, disagree with those of [80], who discovered an upward slope in Ghana's Environmental Kuznets Curve for carbon dioxide emissions, defying the standard Environmental Kuznets Curve theory, which assumes an inverted "U"-shaped relationship between economic growth and environmental degradation. Similar findings were made by [81–84], demonstrating the falsity of the EKC hypothesis.

**Table 8.** Dynamic ARDL simulations analysis.

| | Estimations Using Composite Trade Share (CTS) as a Proxy of Trade Openness | | | Estimations Using Traditional Trade Share (TS) as a Proxy of Trade Openness | | |
| --- | --- | --- | --- | --- | --- | --- |
| | (1) | (2) | (3) | (4) | (5) | (6) |
| Variables | Coefficient | St. Error | *t*-Value | Coefficient | St. Error | *t*-Value |
| Cons | −1.216 | 1.152 | −0.70 | −1.203 | 1.170 | −0.51 |
| lnFD | −0.215 *** | 0.163 | −5.70 | −0.705 ** | 0.173 | −2.45 |
| ΔlnFD | −0.301 ** | 0.804 | −2.46 | −0.583 *** | 0.480 | −4.74 |
| lnGDP | 0.204 *** | 0.171 | 4.72 | 0.173 ** | 0.160 | 2.40 |
| ΔlnGDP | 0.304 *** | 0.208 | 2.83 | 0.410 ** | 0.130 | 2.20 |
| lnGDP$^2$ | −0.617 ** | 0.817 | −2.41 | −0.604 *** | 0.507 | −3.50 |
| ΔlnGDP$^2$ | −0.705 | 0.143 | −1.63 | −0.510 | 0.131 | −1.01 |
| lnOPEN | 0.171 *** | 0.041 | 5.01 | 0.981 ** | 0.610 | 2.51 |
| ΔlnOPEN | −0.141 ** | 0.051 | −2.64 | −0.403 * | 0.130 | −1.99 |
| lnEC | 0.195 *** | 0.160 | 3.13 | 0.240 ** | 0.126 | 2.49 |
| ΔlnEC | 0.511 * | 0.161 | 1.99 | 0.363 ** | 0.102 | 2.48 |
| lnPOP | 1.617 *** | 0.020 | 3.14 | 1.910 | 0.071 | 1.05 |
| ΔlnPOP | 1.242 * | 0.264 | 1.99 | 1.170 *** | 0.504 | 3.81 |
| lnGI | −0.406 *** | 0.411 | −3.01 | 0.801 | 0.133 | 1.40 |
| ΔlnGI | −0.221 | 0.071 | −0.25 | 0.301 | 0.052 | 0.59 |
| lnINS | −0.341 ** | 0.154 | −2.52 | −0.440 * | 0.140 | −1.98 |
| ΔlnINS | −0.593 | 0.227 | −0.10 | −0.410 | 0.211 | −0.25 |
| ECT(−1) | −0.852 *** | 0.136 | −3.14 | −0.714 ** | 0.148 | −2.55 |
| R-squared | 0.898 | | | 0.510 | | |
| Adj R-squared | 0.860 | | | 0.490 | | |
| N | 59 | | | 59 | | |
| P val of F-sta | 0.0000 *** | | | 0.0000 *** | | |
| Simulations | 1000 | | | | | |
| Root MSE | 0.081 | | | 0.271 | | |

Source: Authors' calculations. Note: *, **, and *** denote statistical significance at 10%, 5%, and 1% levels, respectively.

In South Africa, it is observed that the predicted coefficient on fiscal decentralization (lnFD) is negative and statistically significant in both the long and short runs. This empirical evidence demonstrates that in South Africa, a stronger fiscal decentralization may improve the quality of the environment, since local governments operating under the decentralized system have better access to financial resources and more freedom to safeguard the environment. Since local leaders are more knowledgeable about the requirements for local environmental quality, local governments are actually more equipped to spend money on enhancing local environmental quality than the national government. We found proof of the "race to the top strategy", where local South African authorities boost valuations to transfer environmental harm to other jurisdictions. Subnational governments in South Africa have prioritized environmental measures above the federal government in recent decades. The environment is improved through fiscal decentralization, which is a race to the top strategy. Due to the "race to the top" strategy, some regions have implemented strict

environmental regulations and a "beggar-thy-neighbor" policy that allows them to export their damaging practices to surrounding areas. This outcome is consistent with research by [3,4] that demonstrated FD improves environmental quality. When [20] examined the asymmetric impact of fiscal decentralization on $CO_2$ emissions in highly decentralized economies, they arrived at a similar conclusion. The authors also noted that fiscal decentralization employs a number of strategies and institutions that support the restoration of environmental quality, which helps to regenerate the environment. By setting strict environmental standards, fiscal decentralization promotes environmental sustainability and lessens environmental deterioration. Our empirical evidence, however, contradicts the findings of [85], who showed that fiscal decentralization considerably increased $CO_2$ emissions both inside the region and outside of it. Furthermore, ref. [86] noted that in China's economically developed and eastern regions, fiscal decentralization had a detrimental effect on environmental performance.

The calculated coefficient for long-run trade openness (InOPEN) is revealed to be statistically significant and positive, indicating that an increase in trade openness by 1% will, ceteris paribus, result in an increase in $CO_2$ emissions of 0.171%. The long-term negative impact of openness on South Africa's environmental situation unquestionably reinforces the opposition to economic liberalization. Part of the potential explanation for why trade openness harms South Africa's environment is the sort of goods that make up the majority of its exports. Since South Africa has a comparative advantage in the export and production of goods that require a lot of natural resources, such as fuel wood, cerium, metal, base metals, chromium mineral resources, trace elements, lanthanum, rare earth metals, oil and gas, chromite, copper ore, dimes, petroleum products, copper and zinc, gemstones, titanium, and precious metals, an increase in the demand for these goods will undoubtedly worsen the country's environmental situation. This is due to the fact that the constant harvesting of these items to supply the expanding global markets considerably degrades South Africa's environmental quality. Additionally, the [87] theoretical framework, which holds that pollution is mostly caused by energy-intensive activities such as manufacturing and transportation that consume a lot of energy, might be used to explain our findings. Our findings are also in line with the pollution haven hypothesis [88], according to which developing nations such as South Africa have a comparative advantage in producing products that are pollutant-heavy, while developed nations have a comparative advantage in producing products that are clean. Therefore, through international commerce, rich countries frequently transmit pollutants to underdeveloped nations. The findings of [33], which imply that developing nations tend to produce a high quantity of pollutants as a result of reliance on unclean sectors, are consistent with and complement our empirical research. Our findings are in line with those of [17], who claimed that trade openness is detrimental and significantly worsens Pakistan's environmental situation. [31], who concluded that trade openness relates to increased $CO_2$ emissions in the G-7, provide more empirical support for this claim. Similar findings were made by [32], who demonstrated that increased trade openness deteriorates the state of the environment in 66 emerging economies. The findings from [34,89], which revealed that trade has a detrimental influence on environmental quality through releasing carbon dioxide emissions, are consistent with the unfavorable perception of environmental repercussions of trade openness. Our results do not agree with those of [19,28,29,31,82,90], who showed that more trade openness improves environmental quality in the G-20, 48 Sub-Saharan African nations, G-7 economies, and African countries, respectively.

The computed coefficients for the short- and long-term energy consumption (InEC) are statistically significant and positive, indicating that energy consumption significantly contributes to rising $CO_2$ emissions in South Africa. South Africa is the seventh-largest country that largely relies on coal to satisfy its energy needs, and although this is necessary to sustain production and further economic growth, it also significantly contributes to the degradation of environmental quality [12]. It can be shown that a 1% increase in energy use causes a long-term rise in $CO_2$ emissions of 0.195%. South Africa is significantly reliant on

the energy industry, where the production process is dominated by the use of coal. In South Africa, coal reserves account for 93% of power generation and over 77% of the country's primary energy supply [46]. Due to South Africa's consistently rising energy consumption, $CO_2$ emissions have dramatically grown over time, having serious negative effects on the environment, and playing a big role in the global climate change. Ref. [91], who discovered that energy use causes $CO_2$ emissions in South Korea, validate our empirical data. In a similar vein, ref. [92] observed that 17 Mediterranean nations' environmental quality declines as a result of energy usage. According to [93], Turkey's energy use increased $CO_2$ emissions. The same was found by [94], who found that 81 economies participating in the Belt and Road Initiative (BRI) had higher carbon emissions due to increased energy consumption. Research by [95] revealed that in Guangdong, China, overall energy use increased carbon emissions. Our findings conflict with those made by [9,96], who concluded that energy use enhances environmental quality.

In South Africa, a 1% increase in population size (InPOP) is strongly correlated with an increase in $CO_2$ emissions of 1.617% over the long term, according to the calculated coefficient on population size, which is observed to be positive and statistically significant. The correlation between population size and carbon emissions is positive because rising population results in higher energy demands. As a result, there are more fossil fuels burned in the residential sector, which results in more carbon emissions. The findings of [95,97,98] are corroborated by this research. According to the population–emissions elasticity of 1.617, carbon emissions have increased faster than the population. In developing nations more than in industrialized nations, there is a larger prevalence of an elasticity greater than one and, thus, a bigger environmental effect. According to [97], carbon emissions in low-income nations have grown faster than population growth.

The calculated green technological innovation coefficient (InGI) is statistically significant and short-term negative. Our empirical research demonstrates that, over the long term, a 1% increase in green technological innovation results in a 0.406% reduction in $CO_2$ emissions. In South Africa, green technological advancements successfully lower $CO_2$ emissions, which might be viewed as an eco-friendly technological advancement. It encourages effective energy use and makes affordable access to renewable energy sources possible, resulting in South Africa's carbon emissions being kept to a minimum. The following green technological advancements enhance South Africa's environmental quality: (i) installation of end-of-pipe technology essential for lowering carbon emissions; (ii) usage of manufacturing technologies that are more energy efficient; and (iii) adjustments to the fuel mix. Through these channels, technological advancements boost energy efficiency, which significantly enhances the nation's environmental quality. In the meanwhile/, South Africa's considerable R&D expenditures and technological advancements are among the factors that have significantly improved the nation's environmental quality through the use of green technologies. South Africa has also put in place a number of policies to develop strong technologies, which are essential to reducing the intensity of emissions from the production processes and other economic activities linked to high levels of emissions. This is part of the main strategy to mitigate the rising levels of carbon emissions. According to [99], technological innovation opens up the door for lower energy use, greater energy efficiency, and significantly lower carbon emissions. This empirical evidence is supported by [69], who found that green technological innovation contributed to environmental sustainability in South Africa. Our findings further concur with those of [43,100]. The results, however, are at odds with those of [45,101].

It is revealed that the calculated institutional quality coefficient (InINS) is negative and statistically significant. According to our empirical findings, institutional quality in South Africa is inversely connected to $CO_2$ emissions throughout the long and short terms. Strong institutions help to reduce corruption and make it easier to put tough environmental laws into place. In order to reduce climate change and its repercussions through social, governance, and economic preparation, institutional quality is crucial. Therefore, before adaption choices can be implemented, quality political institutions need

several social, governance, and economic changes and policies [48]. Our empirical evidence is in disagreement with [34] but agrees with [102].

The speed of adjustment is captured by the error correction term (ECT). The computed coefficient is negative and statistically significant, indicating that the variables under consideration have a consistent long-term association. According to the ECT calculated value of $-0.85$, 85% of the disequilibrium is likely to be addressed over time. The explanatory factors utilized in this paper account for 89% of fluctuations in $CO_2$ emissions, according to the R-squared value. The model appears to have a decent fit, according to the predicted $p$ value of the F-statistics.

The dynamic ARDL simulations automatically illustrate the predictions of actual regressor change and its influence on the dependent variable while holding other explanatory factors constant. Fiscal decentralization, per capita GDP, per capita GDP squared, trade openness, energy consumption, institutional quality, green technological innovation, and population size are explanatory factors with effects on $CO_2$ emissions that are predicted to fluctuate by 10% in South Africa.

The impulse response plot of South Africa's fiscal decentralization and $CO_2$ emissions is shown in Figure 4. The graph shows that a 10% rise in fiscal decentralization is strongly correlated with both a long-term and short-term negative impact on $CO_2$ emissions. However, a 10% reduction has both a long-term and short-term beneficial impact on $CO_2$ emissions. According to this, increasing fiscal decentralization increases environmental quality, while decreasing it worsens the environment in South Africa over the long and short terms.

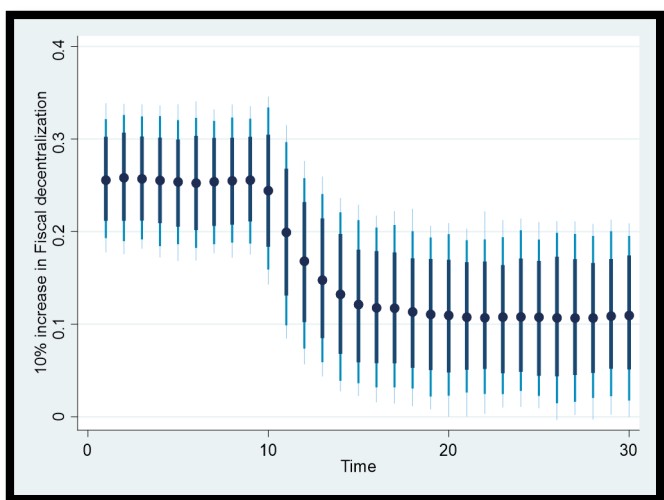 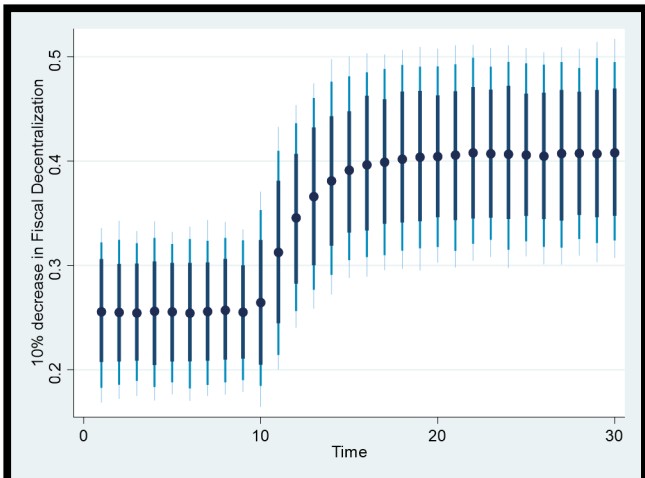

**Figure 4.** A 10% increase and a decrease in fiscal decentralization and its influence on $CO_2$ emissions, where dots specify average prediction value. However, the dark blue to light blue line denotes 75%, 90%, and 95% confidence intervals, respectively.

Figure 5 depicts the link between per capita GDP (economic growth) and $CO_2$ emissions using an impulse-response plot. The plot depicts the change in economic growth and how it affects $CO_2$ emissions. A 10% increase in per capita GDP indicates a long-term and short-term positive impact of economic growth on $CO_2$ emissions, while a 10% decrease in per capita GDP indicates a negative impact of economic growth on $CO_2$ emissions. However, the impact of a 10% increase is greater than that of a 10% decrease in per capita GDP. This suggests that, in South Africa, a rise in economic growth worsens environmental quality, while a reduction in economic growth enhances it over the long and short terms.

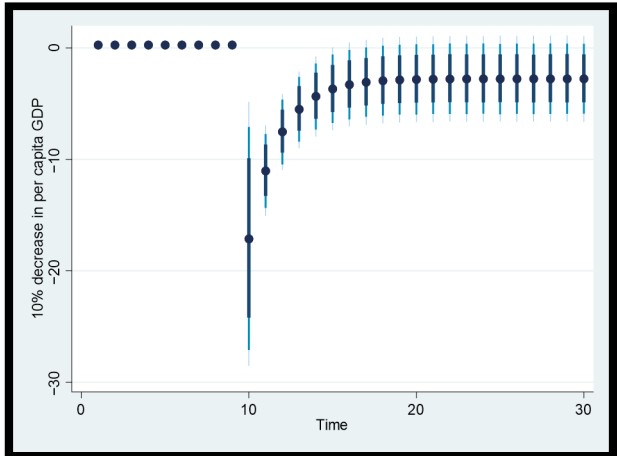 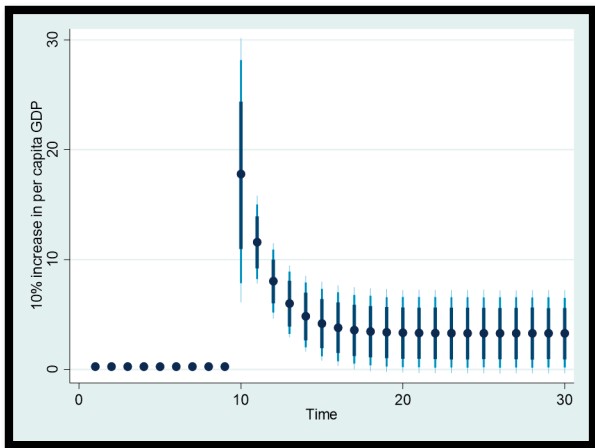

**Figure 5.** The impulse-response plot for per capita GDP (economic growth) and $CO_2$ emissions.

Figure 5 shows a 10% increase and a decrease in per capita GDP and its influence on $CO_2$ emissions, where dots specify average prediction value. However, the dark blue to light blue line denotes 75%, 90%, and 95% confidence intervals, respectively.

The impulse-response plot of South Africa's per capita GDP squared and $CO_2$ emissions is shown in Figure 6. The graph shows that a 10% rise is strongly correlated with a short- and long-term negative impact on $CO_2$ emissions. However, a 10% reduction has both a long-term and short-term positive impact on $CO_2$ emissions. This indicates that in South Africa, a rise in per capita GDP squared enhances environmental quality, but a fall worsens environmental conditions over the long and short terms.

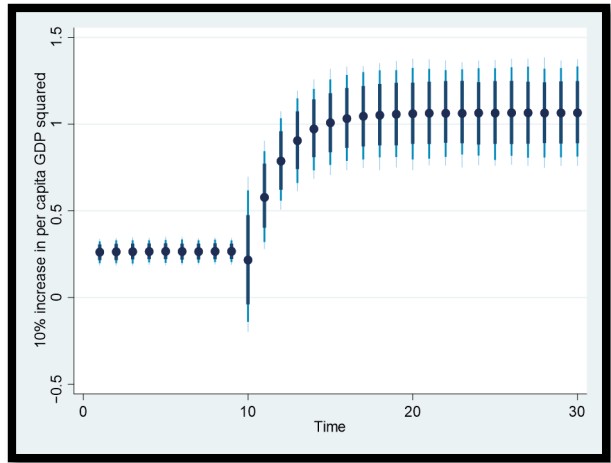 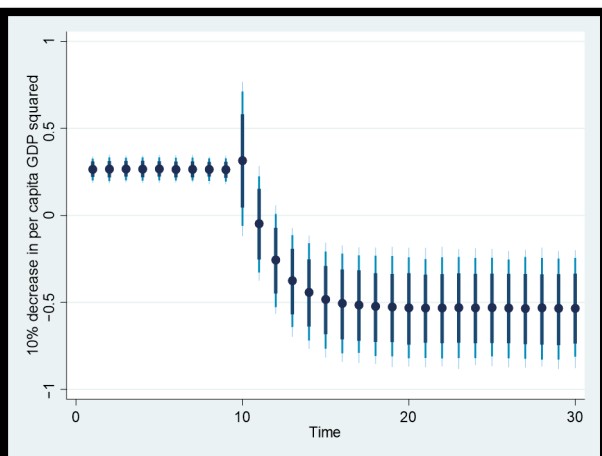

**Figure 6.** The impulse-response plot for per capita GDP squared and $CO_2$ emissions.

Figure 6 shows a 10% increase and a decrease in per capita GDP squared and its influence on $CO_2$ emissions, where dots specify average-prediction value. However, the dark blue to light blue line denotes 75%, 90%, and 95% confidence intervals, respectively.

The impulse-response plot relating trade openness and CO2 emissions is shown in Figure 7. The graph demonstrates how a 10% increase in trade openness has a long-term positive impact on $CO_2$ emissions but a short-term negative impact. In contrast, a 10% reduction in trade openness has a short-term favorable impact on $CO_2$ emissions but a long-term negative impact. This shows that, while a rise in trade openness temporarily enhances South Africa's environmental quality, it really makes things worse over time. However, a decline in trade openness improves South Africa's environment over the long term but worsens it over the short term.

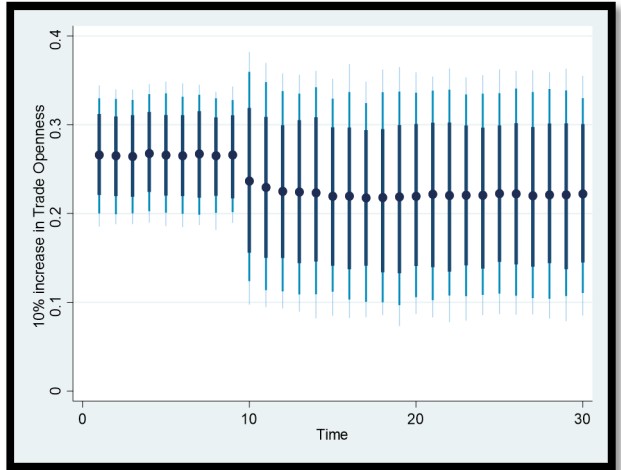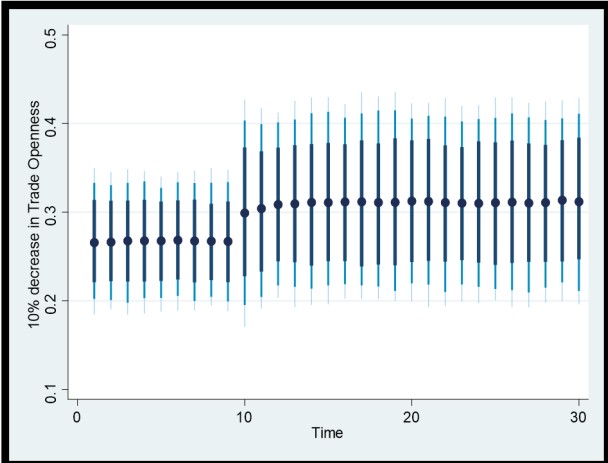

**Figure 7.** The impulse-response plot for trade openness and $CO_2$ emissions.

Figure 7 shows a 10% increase and a decrease in trade openness and its influence on $CO_2$ emissions, where dots specify average prediction value. However, the dark blue to light blue line denotes 75%, 90%, and 95% confidence intervals, respectively.

The impulse response plot illustrating the link between energy use and $CO_2$ emissions is shown in Figure 8. The figure depicting the relationship between energy consumption and $CO_2$ emissions demonstrates that, although a 10% increase in energy use has a positive short- and long-term effect on $CO_2$ emissions, a 10% reduction has the opposite effect. This suggests that, in South Africa, a rise in energy consumption worsens environmental quality, but a decrease in energy use enhances it over the long and short terms.

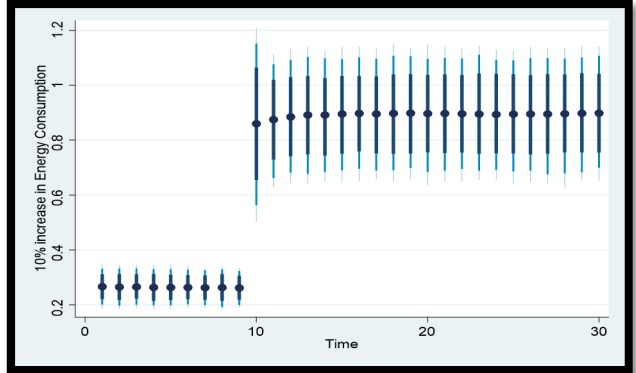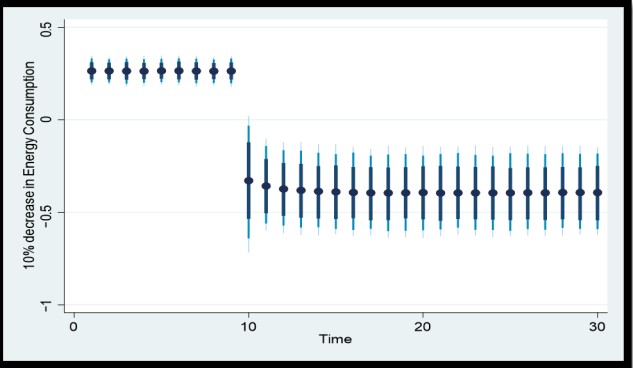

**Figure 8.** The impulse-response plot for energy consumption and $CO_2$ emissions.

Figure 8 shows a 10% increase and a decrease in energy consumption and its influence on $CO_2$ emissions, where dots specify average prediction value. However, the dark blue to light blue line denotes 75%, 90%, and 95% confidence intervals, respectively.

The impulse response plot of South Africa's population and $CO_2$ emissions is shown in Figure 9. The population size graph shows that both a long-term and short-term reduction in $CO_2$ emissions is strongly correlated with a 10% rise in population size. However, a 10% reduction has both a long-term and short-term negative impact on $CO_2$ emissions. This shows that, in South Africa, a rise in population leads to a short- and long-term decline in environmental quality.

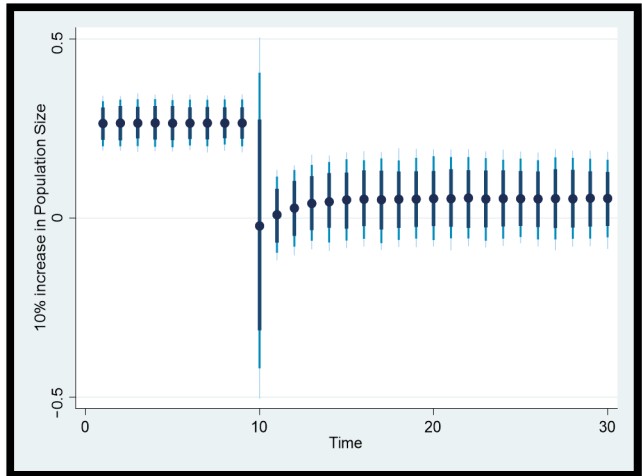
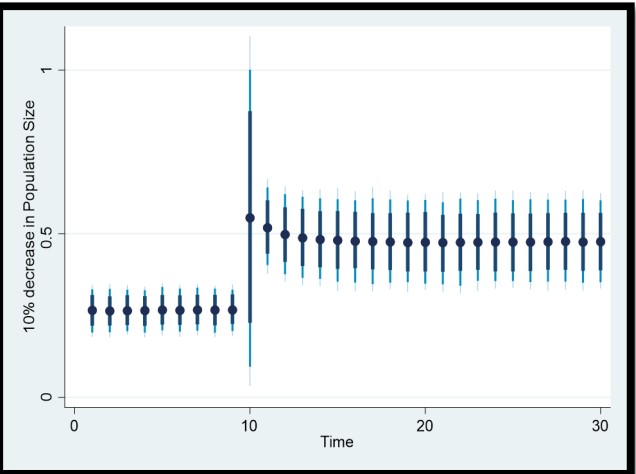

**Figure 9.** The impulse-response plot for population size and $CO_2$ emissions.

Figure 9 shows a 10% increase and a decrease in population size and its influence on $CO_2$ emissions, where dots specify average prediction value. However, the dark blue to light blue line denotes 75%, 90%, and 95% confidence intervals, respectively.

Figure 10 shows the impulse response plot between South Africa's $CO_2$ emissions and advancements in green technology. The graph shows that a 10% increase in green technological innovation has both a long-term and short-term negative impact on $CO_2$ emissions. On the other hand, a 10% decline in technological innovation has an unfavorable long- and short-term impact on $CO_2$ emissions. This shows that a rise in environmentally friendly technological innovation enhances South Africa's environmental quality, but a decline in technological innovation worsens the country's environment over the long and short terms.

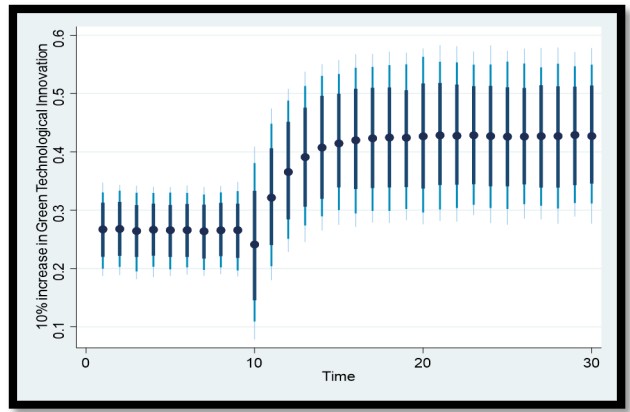
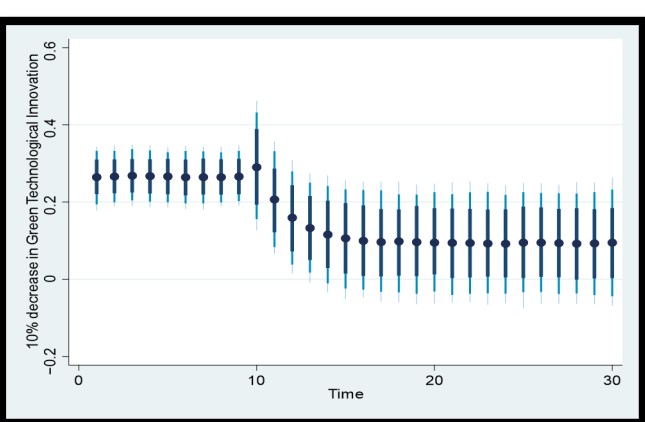

**Figure 10.** The impulse-response plot for green technological innovation and $CO_2$ emissions.

Figure 10 shows a 10% increase and a decrease in green technological innovation and its influence on $CO_2$ emissions, where dots specify average prediction value. However, the dark blue to light blue line denotes 75%, 90%, and 95% confidence interval, respectively.

The impulse response plot showing the link between institutional quality and $CO_2$ emissions is shown in Figure 11. The graph demonstrates that a 10% increase in institutional quality has both short- and long-term advantageous effects on $CO_2$ emissions, whereas a 10% decrease has a detrimental effect. This shows that in South Africa, an improvement in institutional quality helps to enhance environmental quality, but a decline worsens environmental conditions over the short and long terms.

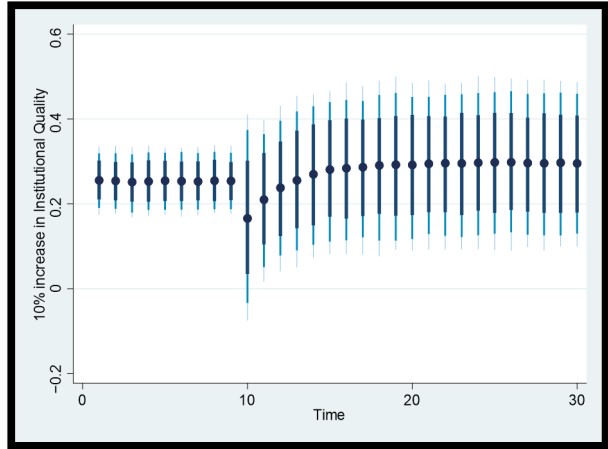 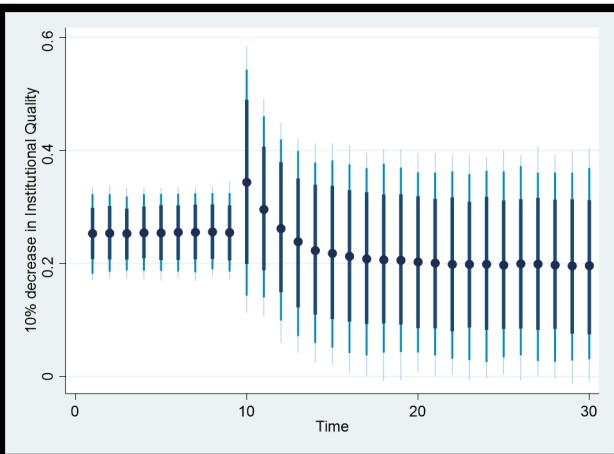

**Figure 11.** The impulse-response plot for institutional quality and $CO_2$ emissions.

Figure 11 shows a 10% increase and a decrease in green technological innovation and its influence on $CO_2$ emissions, where dots specify average prediction value. However, the dark blue to light blue line denotes 75%, 90%, and 95% confidence intervals, respectively.

This study examines the relationship between lnFD, lnGDP lnGDP$^2$, lnOPEN, lnEC, lnPOP, lnGI, lnINS, and lnCO$_2$ in South Africa, using the frequency domain causality test that Breitung and Candelon (2006) developed. For frequencies $\omega_i = 0.05$, $\omega_i = 1.50$, and $\omega_i = 2.50$, Table A1 (see Appendix A) demonstrates that lnFD, lnGDP, lnGDP$^2$, lnOPEN, lnEC, lnPOP, lnGI, lnINS, and lnCO$_2$ Granger-cause lnCO$_2$ in the short, medium, and long term. This suggests that short-, medium-, and long-term $CO_2$ emissions in South Africa are considerably impacted by lnFD, lnGDP, lnGDP$^2$, lnOPEN, lnEC, lnPOP, lnGI, lnINS, and lnCO$_2$. Our empirical evidence agrees with the conclusions reached by [69,73,74,83,103].

To further evaluate its robustness, this work utilizes the structural stability assessment test. To achieve this, ref. [104]'s cumulative sum of recursive residuals (CUSUM) and cumulative sum of squares of recursive residual (CUSUMSQ) are applied. CUSUM and CUSUMSQ are shown graphically in Figures A1 and A2 (see Appendix A). Typically, if plots are within a crucial bound level of 5%, there is a stability of model parameters throughout time. We may infer that the model parameters are consistent over time based on the model trend presented in Figures A1 and A2, and the fact that CUSUM and CUSUMSQ are inside the bounds at a 5% level.

*4.7. Robustness Check*

The first robustness test examines whether the composite trade share (CTS), which is used as a measure of trade openness in this investigation, is superior to the conventional trade share (TS), which was widely used in earlier studies and determines how sensitive the results are to the measurements used. This is accomplished by re-estimating our models, instead of CTS utilizing TS. The findings shown in columns (4)–(6) of Table 8 demonstrate that the explanatory power is significantly reduced when TI is utilized, with the stated $R^2$ value significantly falling from 0.898 to 0.510. Additionally, when adopting the TI metric, the models' root mean squared error (RMSE) considerably rises from 0.081 to 0.271. Furthermore, Table 8's findings demonstrate that using TI instead of CTI significantly increases the predicted coefficient on the relevant variable (fiscal decentralization). More significantly, the size of the disparity between the CTI and TI estimates raises the possibility that utilizing TI might lead to an overestimation of the influence of our variable of interest on South Africa's environmental quality. Additionally, our results are very sensitive when CTI is replaced with TI. The long-run-estimated green technological innovation coefficient (lnGI) is not only statistically significant, but also has the reverse sign, according to the findings. In conclusion, given the aforementioned facts and tests, it is clear that using the CTI measure for model estimation is statistically better, which not only calls into question

the use of the TI measure but also lends statistical support to the use of the CTI measure in this study.

The comparison of the results of the novel dynamic ARDL simulations model with those of the DOLS, FMOLS, and CCR estimation methodologies is the only focus of the second robustness test (see Table A2 in Appendix A). Overall, when the findings were compared, there was little-to-no indication of a difference in the calculated coefficients, particularly when it came to their signs and the degree of statistical significance. While keeping their signs, the majority of the variables are statistically significant, however, occasionally their magnitudes change significantly from one another. These facts lead us to the conclusion that the major findings of the dynamic ARDL simulation model are reliable, consistent, and not significantly different from the findings of the DOLS, FMOLS, and CCR estimation methodologies.

In the last robustness check, since the cointegration findings by Maki cointegration with structural breaks indicated the presence of structural breaks in the data, the research took into account these breaks when estimating the model. We created a dummy variable (D1994) to represent the break year of 1994, which corresponded to the end of apartheid rule and the election of a democratically elected government in South Africa, in order to account for the occurrence of structural breaks in the variables, since only the year 1994 was statistically significant. The findings, which are presented in Table A3 and may be found in Appendix A, suggest that the occurrence of a structural break is not statistically significant.

## 5. Conclusions and Policy Recommendations

### 5.1. Conclusions

Environmental degradation has been a hot topic of debate among economists and environmentalists, while the world confronts the serious issue of global warming, as a result of an increase in climate change. The UN's Sustainable Development Goals (SDGs) have given focus to climate action. Using the newly established novel dynamic ARDL simulations model, proposed by [13] , this investigation explored the dynamic association between fiscal decentralization (FD) and $CO_2$ emissions in the presence of green technological innovation (GI), trade openness (OPEN), population size (POP), per capita GDP (GDP), per capita GDP squared $\left(GDP^2\right)$, institutional quality (INS), and energy consumption (EC) in South Africa over the period of 1960–2020. This method allows us to go beyond the restrictions of the conventional ARDL, by identifying the positive and negative links between FD, GI, OPEN, POP, GDP, $\left(GDP^2\right)$, INS, and $CO_2$ in South Africa. By using a new metric of trade openness developed by [44], which accounts for trade share in GDP and the magnitude of trade relative to global trade for South Africa, this study made an additional contribution to the empirical literature. To verify the asymptotic behavior and order of integration of all variables under consideration, we employed the Dickey–Fuller GLS (DF-GLS), Phillips–Perron (PP), Augmented Dickey–Fuller (ADF), and Kwiatkowski–Phillips–Schmidt–Shin (KPSS) unit root tests. In this way, the problems caused by spurious regressions are addressed by this procedure.

Additionally, Narayan and Popp's structural break unit root test was employed since empirical data indicate that structural breaks are relatively persistent in the empirical literature and that numerous macroeconomic variables, such as $CO_2$ emissions and trade openness, are likely to be impacted. Since there is no indication of any I (2), the empirical evidence from all the tests verified that the data series were integrated into order one, or I (1) (2). In order to determine the ideal lag length, SBIC was used. For a robustness check, we sought to demonstrate that the traditional trade share (TS) extensively employed in past research is inferior to the composite trade share (CTS) utilized as a measure of OPEN in this inquiry. We also examined the degree to which the findings were sensitive to the measurement employed. We also contrasted the results of the novel dynamic ARDL simulation model with those of the DOLS, FMOLS, and CCR estimation methods.

Since the cointegration findings from Maki's cointegration with structural breaks highlighted the existence of structural breaks in the data, the research took these breaks into account while estimating the model. Since only the year 1994 was statistically significant, we created a dummy variable (D1994) to stand in for the break year, which was the year when apartheid rule in South Africa came to an end and a democratically elected government was chosen. The Breitung and Candelon (2006) robust testing technique of frequency domain causality (FDC) was employed to allow us to capture permanent causation for medium-, short-, and long-term effects.

For South Africa, our empirical results revealed that (i) FD, GI, and INS improved environmental sustainability in both the short and long run; (ii) OPEN deteriorated environmental quality in the long run, although it is environmentally friendly in the short run; (iii) per capita GDP increased $CO_2$ emissions, whereas its square contributed to lower it, thus validating the presence of an environmental Kuznets curve (EKC) hypothesis; (iv) POP and EC contributed to environmental deterioration in both the short and long run; and (v) FD, GI, OPEN, POP, GDP, $GDP^2$, INS, and EC Granger-caused $CO_2$ in the medium, long, and short run, suggesting that these variables are important to influence environmental sustainability.

### 5.2. Policy Implications

We propose the following policy suggestions based on our empirical findings:

Firstly, the results of this study can aid in the development of environmental and FD policies in South Africa. South Africa should create legislation to reduce emission levels to control the worsening environmental quality. Similar to other nations, the promotion of energy-efficient systems is essential to converting the industrial structure to renewable energies. In order to realize the goal of low $CO_2$ emissions and energy-saving fiscal-spending functions, it is also crucial to clearly define the roles and duties of the various levels of government. It is necessary to concentrate on environmentally friendly technologies that transfer the economic development drivers from non-renewable sources to long-term resources such as renewable energy. Ecology and climate change will be greatly affected by these eco-friendly technologies. In order to develop GI, South Africa should also change its industrial and economic structure.

Second, since GI is a crucial component of achieving the Sustainable Development Goals (SDGs), policymakers should strengthen policies that encourage GI activities and investment, which promote growth while also further reducing emissions. This is because our empirical findings demonstrate that GI contributes to improving South Africa's environmental sustainability. A green-innovation aid program can be formed to advance green technological advances by combining financial and material resources. Governmental research projects and commercial and public organizations' green research initiatives may be supported by government funding. Furthermore, GI may be enhanced by using green financing. It is beneficial to use carbon sequestration technology in power plants and to apply GI to promote green investments.

Thirdly, the information at hand points to an advantageous role for institutional excellence in environmental sustainability. South Africa should, thus, strengthen its ability to manage and keep establishing top-notch institutions to organize and control frameworks for sustainable growth. On the other side, robust institutions can efficiently control financial institution-related activities, slowing the expansion of polluting projects.

Fourthly, in order to address the rising trans-boundary environmental deterioration and other related spillover effects, international cooperation to improve environmental quality is very essential. In light of this, the government should collaborate to create strong international relationships with other nations in order to share technologies.

Lastly, to facilitate a transition to a low-carbon economy and greener businesses, the government should incorporate comprehensive environmental chapters into the nation's trade agreement rules. This would encourage the creation of cleaner products. To assure long-term value for carbon emission reductions and consistently encourage the development of new technologies that can improve the nation's environmental quality

and safeguard the global environment, trade policy reform might be backed by other developmental policies.

*5.3. Limitations and Potential Future Research Areas*

Although this paper investigated the impact of FD on $CO_2$ emissions in the presence of some control variables such as GI, OPEN, POP, GDP, $GDP^2$, INS, and EC, there are some areas that need to be improved in future research. First, we found that FD, GI, OPEN, POP, GDP, $GDP^2$, INS, and EC are all important factors affecting environmental sustainability in South Africa. The results of this study can be extended to other countries, regions, and groups. Second, the scope of this study is limited to South Africa, and only a limited number of variables are included. Moreover, the time dimension of this study is limited from 1960 to 2020. Future studies can extend the model by incorporating other variables and conducting a panel data analysis of African countries and other emerging nations. Third, the measurement model of FD will be improved to enhance measurement accuracy. Fourth, the impact mechanism of FD on $CO_2$ emissions will be explored. Fifth, extending the research sample to more periods would be meaningful. Lastly, future research is required to identify the complementarities of FD in affecting $CO_2$ emissions.

**Author Contributions:** Conceptualization, M.C.U. and N.N.; Data curation, M.C.U. and N.N.; Formal analysis, M.C.U. and N.N.; Investigation, M.C.U. and N.N.; Methodology, M.C.U. and N.N.; Resources, M.C.U. and N.N.; Software, M.C.U. and N.N.; Supervision, N.N.; Validation, M.C.U. and N.N.; Visualization, M.C.U. and N.N.; Writing—original draft, M.C.U. and N.N.; Writing—review & editing, M.C.U. and N.N. All authors have read and agreed to the published version of the manuscript.

**Funding:** This research received no external funding.

**Institutional Review Board Statement:** Not applicable.

**Informed Consent Statement:** Not applicable.

**Data Availability Statement:** The dataset generated during and/or analysed during the current study are publicly available from the World Bank's World Development Indicators.

**Conflicts of Interest:** The authors declare no conflict of interest.

## Appendix A

**Table A1.** Frequency-domain causality test $\omega_i = 1.50$.

| Direction of Causality | Long-Term | Medium-Term | Short-Term |
|---|---|---|---|
| | $\omega_{i=0.05}$ | $\omega_{i=1.50}$ | $\omega_{i=2.50}$ |
| $lnGDP \rightarrow lnCO_2$ | <9.72> | <8.81> | <9.45> |
| | (0.00) *** | (0.00) *** | (0.03) ** |
| $lnGDP^2 \rightarrow lnCO_2$ | <4.53> | <6.31> | <6.01> |
| | (0.06) * | (0.02) ** | (0.02) ** |
| $lnFD \rightarrow lnCO_2$ | <9.76> | <8.06> | <7.35> |
| | (0.00) *** | (0.00) *** | (0.00) *** |
| $lnGI \rightarrow lnCO_2$ | <4.19> | <6.03> | <6.46> |
| | (0.07) * | (0.03) ** | (0.04) ** |
| $lnEC \rightarrow lnCO_2$ | <4.42> | <7.68> | <5.40> |
| | (0.07) * | (0.00) *** | (0.04) ** |
| $lnPOP \rightarrow lnCO_2$ | <5.10> | <6.25> | <7.31> |
| | (0.08) * | (0.02) ** | (0.00) *** |
| $lnINS \rightarrow lnCO_2$ | <5.16> | <7.05> | <7.46> |
| | (0.00) *** | (0.03) ** | (0.04) ** |
| $lnOPEN \rightarrow lnCO_2$ | <5.51> | <8.63> | <8.71> |
| | (0.03) ** | (0.00) *** | (0.00) *** |

Source: Authors' calculations. Note: *, **, and *** denote statistical significance at 10%, 5%, and 1% levels, respectively.

**Table A2.** Long-run estimates.

| Variables | Coefficients$_{CCR}$ [Standard Error] | Coefficients$_{FMOLS}$ [Standard Error] | Coefficients$_{DOLS}$ [Standard Error] |
|---|---|---|---|
| lnFD | −0.320 *** [0.073] | −0.371 *** [0.085] | −0.314 *** [0.031] |
| lnGDP | 0.228 *** [0.021] | 0.160 [0.517] | 0.136 *** [0.036] |
| lnGDP$^2$ | −0.502 ** [0.058] | −0.540 ** [0.043] | −0.517 *** [0.072] |
| lnOPEN | 0.153 ** [0.016] | 0.107 *** [0.021] | 0.140 [0.598] |
| lnEC | 0.190 *** [0.072] | 0.195 ** [0.048] | 0.189 [0.495] |
| lnGI | −0.205 * [0.164] | −0.259 ** [0.152] | −0.241 ** [0.160] |
| lnPOP | 0.791 *** [0.140] | 0.795 ** [0.016] | 0.694 [0.318] |
| lnINS | −0.162 *** [0.061] | −0.157 *** [0.068] | −0.165 *** [0.071] |
| Constant | −0.281 ** [0.013] | −0.263 *** [0.048] | −0.201 ** [0.031] |

Note: ***, **, and * denote 1%, 5%, and 10% significance levels, respectively. FMOLS: fully modified ordinary least squares; DOLS: dynamic ordinary least squares; CCR: canonical cointegrating regression.

**Table A3.** Dynamic ARDL simulations analysis controlling for structural break.

| | Estimations Using Composite Trade Intensity (CTI) as a Proxy of Trade Openness | | |
|---|---|---|---|
| | **(1)** | **(2)** | **(3)** |
| Variables | Coefficient | St. Error | *t*-value |
| Cons | −1.2162 | 1.1524 | −0.70 |
| D94 | 0.0253 | 0.1703 | 0.51 |
| lnFD | −0.2102 *** | 0.1830 | −4.72 |
| ΔlnFD | −0.2518 ** | 0.8313 | −2.51 |
| ln GDP | 0.1942 *** | 0.2010 | 3.72 |
| ΔlnGDP | 0.3121 *** | 0.2151 | 2.74 |
| lnGDP$^2$ | −0.6181 ** | 0.8143 | −2.50 |
| ΔlnGDP$^2$ | −0.7036 | 0.1416 | −1.06 |
| lnOPEN | 0.1225 *** | 0.0442 | 4.05 |
| ΔlnOPEN | −0.2345 ** | 0.0529 | −2.74 |
| lnEC | 0.1957 *** | 0.2003 | 3.41 |
| ΔlnEC | 0.4814 * | 0.1615 | 1.99 |
| lnPOP | 0.6171 | 0.0804 | 1.14 |
| ΔlnPOP | 0.2421 ** | 0.2641 | 2.48 |
| lnGI | −0.4066 *** | 0.4117 | −3.01 |
| ΔlnGI | −0.2210 | 0.0715 | −0.25 |
| lnINS | −0.3414 ** | 0.1542 | −2.52 |
| ΔlnINS | −0.5935 | 0.2270 | −0.10 |
| ECT(−1) | −0.8521 *** | 0.1364 | −3.14 |
| R-squared | 0.898 | | |
| Adj R-squared | 0.860 | | |
| N | 59 | | |
| P val of F-sta | 0.0000 *** | | |
| Simulations | 1000 | | |
| Root MSE | 0.081 | | |

Source: Authors' calculations. Note: *, **, and *** denote statistical significance at 10%, 5%, and 1% levels, respectively.

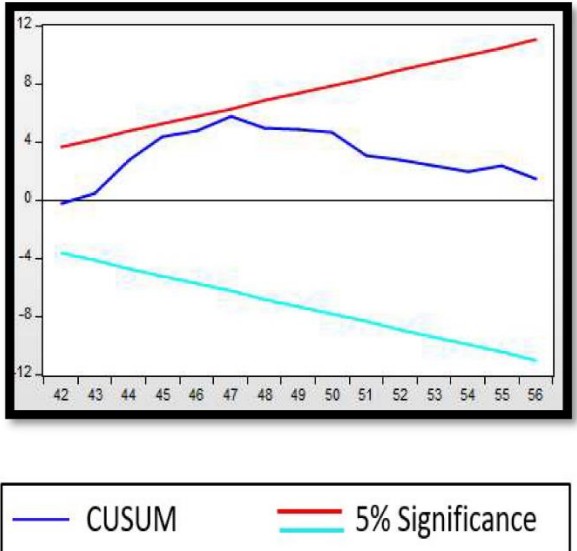

**Figure A1.** Plot of cumulative sum of recursive residuals (CUSUM).

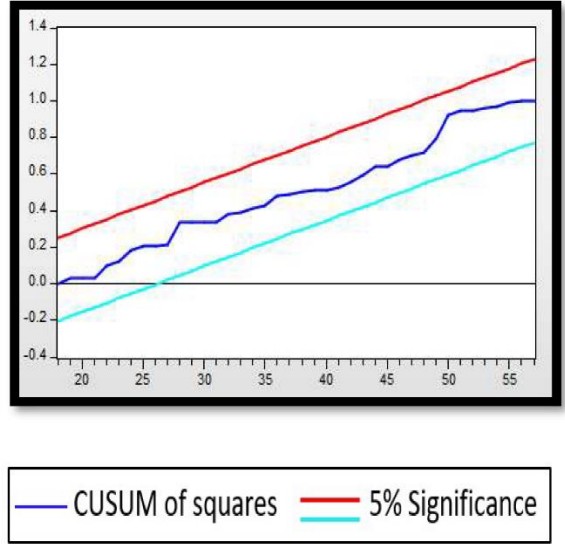

**Figure A2.** Plot of cumulative sum of squares of recursive residuals (CUSUMSQ).

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
