# Peer review of "Dynamic ARDL Simulations Effects of Fiscal Decentralization, Green Technological Innovation, Trade Openness, and Institutional Quality on Environmental Sustainability: Evidence from South Africa"

_sustainability, doi:10.3390/su141610268_

Round 1
Reviewer 1 Report
The paper abord the environmental sustainability analize based on economic point of view and explore the effect of fiscal decentralization and technological innovation in South Africa in the context of globalization and GDP.
As sugestion, the paper must be improve about the empiracal model presented in section 3.1. and detailed explanations about the variables taking in account is required. Maybe, an experimental analysis study can help to determine the variables sensibility and to explain the interrelation between them.
About the results and discussion section must be show how the variables evolution could explain the CO2 mitigation from FD, GDP, and so on.
As additional question, the authors could be explain the inpact on energy-efficient system over the CO2 mitigation as they comment in the abstract paper.
Finally, the paper presentation could be improve.
Reviewer 2 Report
Journal: Sustainability
Manuscript ID: sustainability-1790584
Title: Discussing the role of fiscal decentralization and technological 2 innovation in environmental sustainability in South Africa
The manuscript titled “Discussing the role of fiscal decentralization and technological 2 innovation in environmental sustainability in South Africa” is well-structured and well-written. The review findings of the work will be useful to the readers of the Jounal.However, following suggestions are provided to highlight the key scientific merit of the work.
1.The abstract needs modification. Too short and lack of information. Please expand and modify the abstract with background, introduction, methodology, results and conclusion
2. Introduction section must be written on more quality way, The various economic activities and other factors contributing to the environmental sustainability has to be highlighted. More up-to-date references addressed. Research gap should be delivered on more clear way with directed necessity for the conducted research work on environmental innovation and policies. Some of the references cited in the introduction are outdated. It is suggested to add 2020-22 references. The novelty and research gap of the study are not clearly identified in the work. The fundamental problem to be solved is not clearly identified.
3. The objective and methodology used in the work should be clearly stated. The Fiscal decentralization (FD), TI, Gland GDP influence on CO2 emission needs to be presented. The purpose and procedure of environmental sustainability needs to be provided. The Co-integration test, unit root-analysis , Frequency-domain causality test details are to be provided The novelty, research gap and the fundamental problem to be solved is not clearly identified in the work.
4. A good research work should contain a clear methodology and explanation of the key terms. Please improve the Methods section. The empirical and theatrical model assumptions are very shallow. Please improve these sections
5. The manuscript contains lot of Typographical, alignment and grammatical errors:. Please check the whole manuscript.
6. The analysis and results section have to be elaborated. The globalization, GDP, fiscal decentralization, eco-innovation, economic growth, and international trade may be discussed and presented in a tabular format.
7. A flowchart may be included to present the whole methodology if the work.
8. A good research work should contain good supporting figures or images. please add more suitable figures if possible. Please discuss the results presented in various tables (Table 1,2 and 3)
9. The conclusion section is weak with little or no numbers to support the findings. Please include the limitations, future scope and practical implication of the study
Reviewer 3 Report
This paper analyses the effect of fiscal decentralization and technological innovation on ecological quality in South Africa from 1960 to 2020 in the context of globalization and GDP
The authors consider that there are many factors that influence on CO2 emissions but that fiscal decentralization and technological innovation have not been enough studied. That is why they have interest in these topic. I think that is not enough.
1. Authors have to justify better why they dis this analysis
2. They have to justify why they include in addition GDP and globalization and not other variables, as renewable, urbanization, trade index....
3. Why they have not consider GDP squared, as in many previous studies
In the methodological approach, They have to justify why they have not include a time trend variable or similar, ...
Why they use FMOLS, DOLS...instead of other methods,please justify better
Round 2
Reviewer 3 Report
I think the paper has been improved and can be accepted for publication